# BiteOscope, an open platform to study mosquito biting behavior

Felix JH Hol[1,2,3]*, Louis Lambrechts[2], Manu Prakash[1]

[1]Department of Bioengineering, Stanford University, Stanford, United States; [2]Insect-Virus Interactions Unit, Institut Pasteur, UMR2000, CNRS, Paris, France; [3]Center for research and Interdisciplinary, U1284 INSERM, Université de Paris, Paris, France

**Abstract** Female mosquitoes need a blood meal to reproduce, and in obtaining this essential nutrient they transmit deadly pathogens. Although crucial for the spread of mosquito-borne diseases, blood feeding remains poorly understood due to technological limitations. Indeed, studies often expose human subjects to assess biting behavior. Here, we present the biteOscope, a device that attracts mosquitoes to a host mimic which they bite to obtain an artificial blood meal. The host mimic is transparent, allowing high-resolution imaging of the feeding mosquito. Using machine learning, we extract detailed behavioral statistics describing the locomotion, pose, biting, and feeding dynamics of *Aedes aegypti, Aedes albopictus, Anopheles stephensi,* and *Anopheles coluzzii.* In addition to characterizing behavioral patterns, we discover that the common insect repellent DEET repels *Anopheles coluzzii* upon contact with their legs. The biteOscope provides a new perspective on mosquito blood feeding, enabling the high-throughput quantitative characterization of this lethal behavior.

## Introduction

Blood feeding is essential for the reproduction of many mosquito species, and in the process, mosquitoes transmit myriad pathogens to their (human) host. Yet, despite being the focal point of pathogen transmission, many aspects of blood feeding remain ill understood. The initial step in obtaining a blood meal, flying toward a host, is relatively well characterized (*Dekker and Cardé, 2011*; *McMeniman et al., 2014*; *van Breugel et al., 2015*). The steps that unfold after a mosquito has landed on a host, however, are much less understood. Once landed, mosquitoes exhibit exploratory bouts during which the legs and proboscis frequently contact the skin (*Jones and Pilitt, 1973*; *De Jong and Knols, 1995*; *Clements, 2013*). An increasing body of literature reports the presence of receptors involved in contact-dependent sensing on the legs and proboscis (*Sparks et al., 2013*; *Matthews et al., 2019*; *Dennis et al., 2019*), suggesting that these appendages evaluate the skin surface and thus serve an important role in bite-site selection. Yet, the role and mechanism of contact-dependent sensing in blood feeding is largely unclear (*Benton, 2017*). In addition to the body parts that come in contact with the skin surface, the skin piercing labrum also serves as a chemosensory organ, guiding blood feeding in currently unknown ways (*Lee, 1974*; *Werner-Reiss et al., 1999*; *Jove et al., 2020*).

In addition to external cues, an animal's (internal) physiology may also affect its behavior. Nutrition, hydration, and pathogen infections, for instance, have been hypothesized to affect blood feeding behavior, for example by altering feeding avidity (i.e. number of feeding attempts) or the size of the meal taken (*Rossignol et al., 1984*; *Choumet et al., 2012*; *Cator et al., 2013*; *Vantaux et al., 2015*; *Hagan et al., 2018*). These topics, however, remain a matter of debate, due to a lack of (standardized) assays to measure mosquito behavior (*Stanczyk et al., 2017*). Quantitative mapping of *Drosophila* behavior provides an important perspective, suggesting that innovative experimental

*For correspondence:
felix.hol@pasteur.fr

**Competing interests:** The authors declare that no competing interests exist.

**eLife digest** Scientists often sacrifice their own skin to study how mosquitos drink blood. They allow mosquitos to bite them in laboratory settings so they can observe the insects' feeding behavior. By observing blood feeding, scientists hope to find ways to prevent deadly diseases like malaria, which is transmitted by bites from mosquitos carrying the malaria parasite. These studies are not only unpleasant for the volunteers, they also have important limitations. For example, it is too risky to use pathogen-infected mosquitos that could make the volunteers sick.

A device called the biteOscope developed by Hol et al. may give scientists and their skin a reprieve. The device has a transparent skin-like covering that attracts mosquitos and supplies them an artificial blood meal when they bite. The device captures high-resolution images of the insects' behavior. It is small enough to fit in a backpack when disassembled, costs about $900 to $3,500 US dollars, and is suitable for use in the laboratory or in the field. Using machine-learning techniques, Hol et al. also developed an automated system for analyzing the images.

The researchers tested the device on four types of disease-transmitting mosquitos. In one set of experiments, *Anopheles* mosquitos were recorded interacting with a biteOscope partially coated with an insect repellent called DEET. The images captured by the biteOscope showed that the mosquitos are attracted to the warm surface and land on the part coated with DEET. But when their legs come in contact with the repellent, they leave.

The biteOscope provides scientists a new way to study blood feeding, even in mosquitos infected with dangerous pathogens. It might also be used to test new ways to prevent mosquitos from biting and spreading disease. Because the device is portable and relatively inexpensive, it may enable larger studies in a variety of settings.

approaches and computational tools can fuel the acquisition of new insights (e.g. *Branson et al., 2009*; *Kain et al., 2013*; *Berman et al., 2014*; *Corrales-Carvajal et al., 2016*; *Robie et al., 2017*; *Moreira et al., 2019*). Yet, apart from olfactometers and other flight chambers, very few assays to characterize the blood-feeding behavior of mosquitoes exist (*Geier and Boeckh, 1999*; *Verhulst et al., 2011*; *McMeniman et al., 2014*; *van Breugel et al., 2015*; *Murray et al., 2020*). Due to this paucity of assays, studies often expose human subjects to quantify the number of landings and/or bites, or the time it takes to complete a blood meal, and score experimental outcomes by hand (*Jones and Pilitt, 1973*; *Ribeiro, 2000*; *Moreira et al., 2009*; *DeGennaro et al., 2013*; *Dennis et al., 2019*; *Hughes et al., 2020*). The use of humans as bait constrains the number and type of experiments that can be done (e.g. prohibiting the use of infected mosquitoes) and limits the type, detail, and throughput of measurements that can be made. Furthermore, the opaque nature of skin prevents the visualization of the stylets after piercing the skin leaving this aspect of blood feeding almost entirely unstudied, except for one notable study using intravital imaging of dissected mouse skin (*Choumet et al., 2012*) and two much earlier descriptions (*Gordon and Lumsden, 1939*; *Griffiths and Gordon, 1952*).

To overcome these limitations, we developed the biteOscope, an open platform that allows the high-resolution and high-throughput characterization of surface exploration, probing, and engorgement by blood-feeding mosquitoes. The biteOscope consists of a rudimentary skin mimic: a substrate that attracts mosquitoes to its surface, induces them to land, pierce the surface, and engage in blood feeding. The bite substrate can be mounted in the wall of a mosquito cage allowing freely behaving mosquitoes access. By virtue of its transparent nature, the substrate facilitates imaging of mosquitoes interacting with it, including the visualization of the skin piercing mouthparts of the mosquito. We developed a suite of computational tools that automates the extraction of behavioral statistics from image sequences, and use machine learning to track the individual body parts of behaving mosquitoes. These capabilities enable a detailed characterization of blood-feeding mosquitoes. We demonstrate that the biteOscope is an effective instrument to study the behavior of several medically relevant species of mosquito and describe behavioral patterns of the two main vectors of dengue, Zika, and chikungunya virus (*Aedes aegypti* and *Aedes albopictus*), and two important malaria vectors (*Anopheles coluzzii* and *Anopheles stephensi*). The biteOscope allows detailed tracking of the complex interactions of mosquitoes with a substrate and can be used to characterize

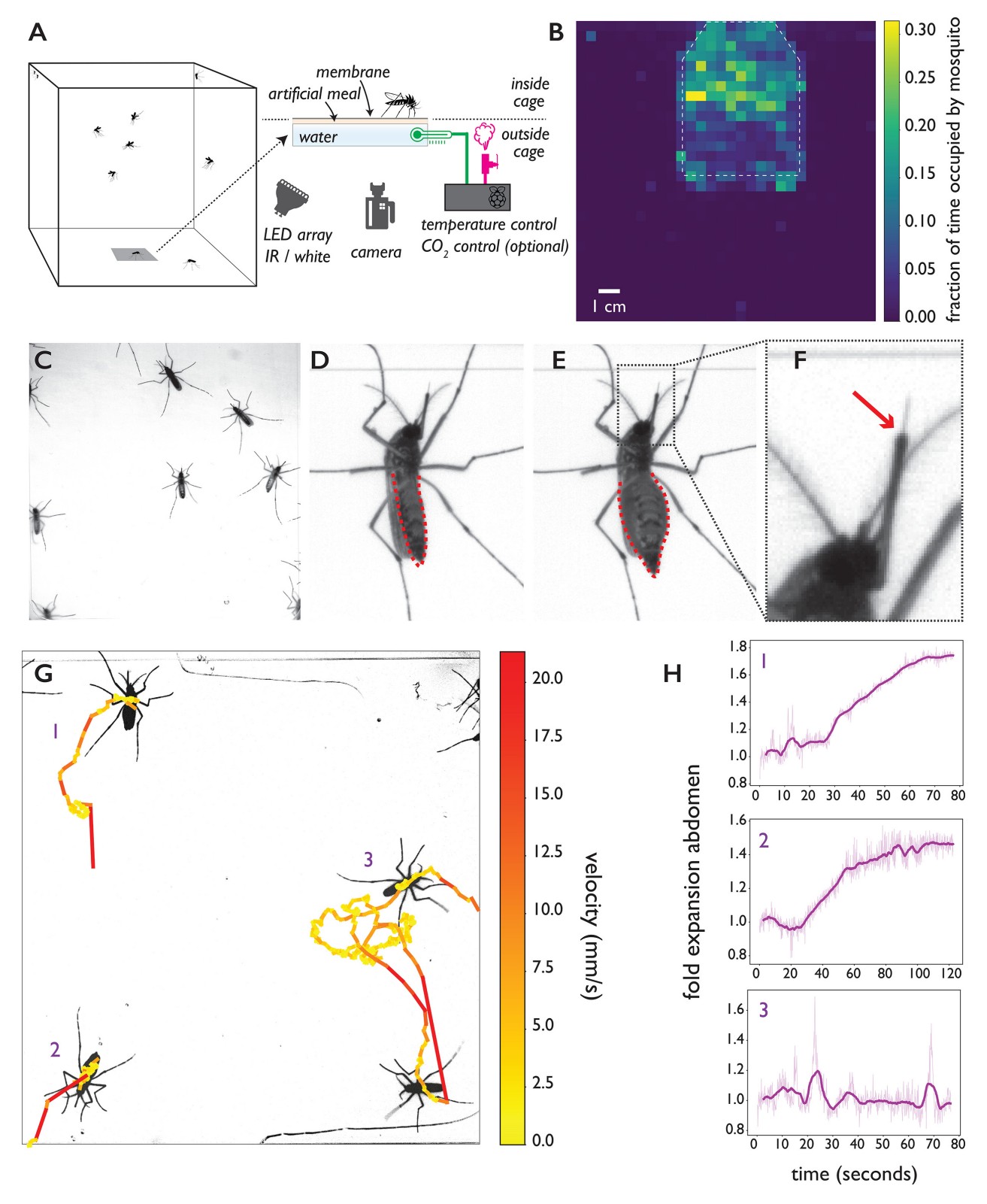

**Figure 1.** The biteOscope. (**A**) Schematic of the set up. The bite substrate consists of a water bath (cell culture flask) that is mounted in the floor or wall of a cage, allowing freely flying mosquitoes access. An artificial meal is applied on the outside surface of the culture flask and covered using a Parafilm membrane, water in the flask is temperature controlled using a Raspberry Pi reading a temperature probe, and a Peltier element for heating (0.1 accuracy). The Raspberry Pi optionally controls the inflow of gas. Illumination is provided by an array of white or IR LEDs. A camera and lens situated

*Figure 1 continued on next page*

*Figure 1 continued*

outside the cage images mosquitoes (abdominal view) through the bite substrate. (B) Two-dimensional histogram (heatmap) showing mosquito presence on the bite substrate (indicated with a dashed line) and on the surrounding wall. Mosquitoes spend more time on the bite surface. (C) Raw image of *Ae. aegypti* on the bite substrate. (D-F) Images of an *Ae. aegypti* mosquito that has pierced the membrane and inserted its stylet into the meal. After imbibing, the abdomen dilates. The red arrow in (F) indicates the tip of the labium where the stylets (visible as a thin needle-like structure) pierce the surface and enter the artificial meal. (G) Tracks showing movement of *Ae. aegypti* on the bite substrate, color of tracks indicates velocity. (H) Fold expansion of the abdomen over time, indicating full engorgement in mosquitoes 1 and 2, and no feeding in mosquito 3 of panel (G). The online version of this article includes the following figure supplement(s) for figure 1:

**Figure supplement 1.** Schematic of bite substrate assembly.
**Figure supplement 2.** Overview of the computational pipeline.

behavioral alterations in the presence of chemical surface patterns. Using this capability, we provide evidence that DEET repels *Anopheles coluzzii* upon contact with their legs, demonstrating the utility of body part tracking to understand behaviors mediated by contact-dependent sensing. We anticipate that the biteOscope will enable studies that increase our understanding of the sensory biology and genetics of blood feeding, and the effects external (environmental) and internal (physiology) variables have on this behavior. Given its relevance for pathogen transmission, dissecting the interplay between the mosquito sensory system and host-associated cues during blood feeding is of clear interest, and may suggest new avenues to interfere with blood feeding, and eventually curb pathogen transmission.

## Results

### The biteOscope

To allow mosquitoes to engage in blood feeding and feed to full repletion, a device needs to attract mosquitoes, allow them to explore and

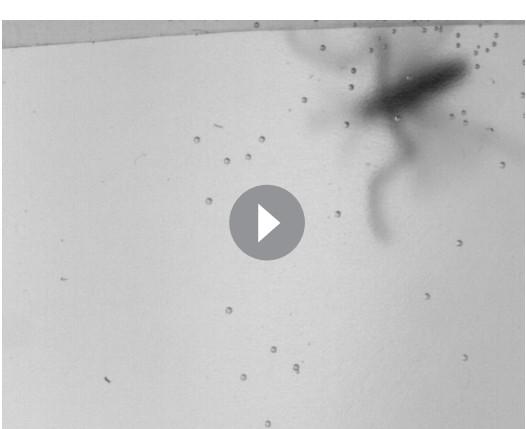

**Video 1.** *Ae. albopictus* female landing, probing, and feeding to full repletion. Upon landing, the mosquito walks/explores the substrate for a short period to pierce the surface and insert her stylets, clearly visible as a flexible needle. The video shows a fast pulling motion of the fore and hind legs towards the body which is typical during the probing phase. While engorging, the body remains nearly motionless and the abdomen dilates visibly.
https://elifesciences.org/articles/56829#video1

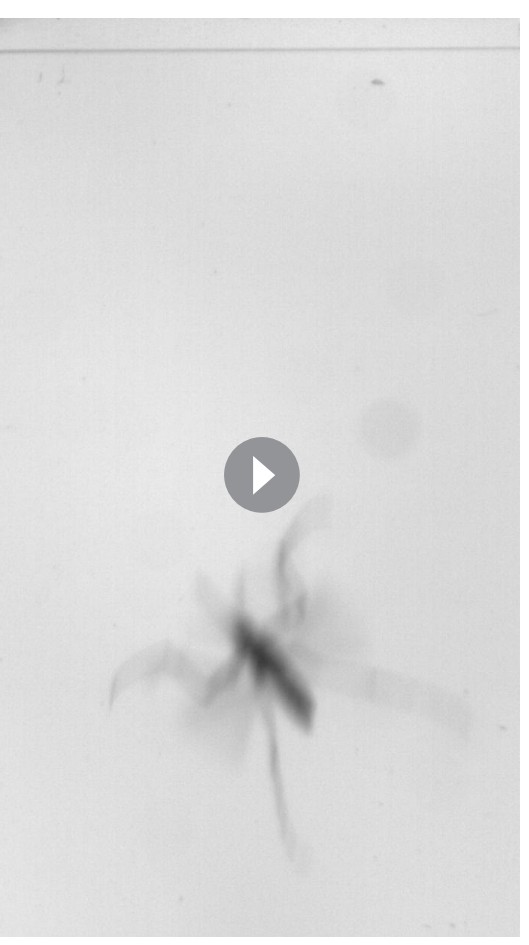

**Video 2.** An *Ae. aegypti* female lands, probes (visible as a pulling motion towards the body), walks several millimeters, probes again, and finally starts to engorge. Engorgement is clearly visible as a dilation of the abdomen. Video playing in real time.
https://elifesciences.org/articles/56829#video2

pierce the surface, and subsequently imbibe a blood meal. To design a tool that can easily be used in a variety of 'mosquito labs' (including (semi-)field settings), we sought to recapitulate this behavioral sequence using readily available and low-cost laboratory materials. Heat is a dominant factor in short-range mosquito attraction and can be used to attract mosquitoes to a surface and elicit probing behavior (*Healy et al., 2002*; *Corfas and Vosshall, 2015*; *Zermoglio et al., 2017*; *Greppi et al., 2020*). We constructed a bite substrate using an optically clear flask filled with water as a controllable heat source (see *Figure 1A*). An artificial blood meal is applied on the outside of the flask and covered using Parafilm (a commonly used membrane in laboratory blood feeders) creating a thin fluid cell on which mosquitoes can feed (see *Figure 1—figure supplement 1*). To elicit blood feeding in a transparent medium, we use adenosine triphosphate (ATP) as a strong phagostimulant, which, together with an osmotic pressure similar to that of blood and the presence of sodium ions, is sufficient to induce *Aedes* mosquitoes to feed to full engorgement (*Galun et al., 1963*; *Duvall et al., 2019*). *Anopheles* also require sodium ions and a tonicity similar to blood to feed to full engorgement, but interestingly their feeding rate on artificial meals is independent of ATP (*Galun et al., 1985*).

To allow freely behaving mosquitoes access to the bite substrate, we constructed acrylic cages having an opening in the wall or floor where the bite substrate can be mounted. The bite substrate is transparent, facilitating imaging with a camera mounted outside the cage (*Figure 1A* shows a schematic of the set up). For the majority of data presented here, we used a 4.3 × 4.3 cm field of view (see *Figure 1C*) which allows up to 15 mosquitoes to explore and feed simultaneously while providing images at a resolution where small body parts like the stylets can easily be resolved. Depending on experimental requirements, the field of view (and correspondingly assay throughput) can be much larger at the expense of resolution. *Figure 1B*, for example, shows a 13 × 13 cm field of view. Individual mosquitoes can be easily tracked at that resolution, yet the visualization of small body parts is challenging. Experiments on *Ae. aegypti* and *Ae. albopictus*, both active during the day, were performed using white light illumination; we used an infrared (IR) LED array as light source during experiments on *An. coluzzii* and *An. stephensi* which were performed in the dark, corresponding to their peak activity during the night. *Figure 1B* demonstrates that *Ae. aegypti* mosquitoes show strong attraction to the bite substrate (surface indicated using a dashed line) and spend more time on its surface compared to the surrounding wall. *Figure 1C–F* shows *Ae. aegypti* undertaking the full blood feeding trajectory on the substrate: starting with surface exploration (*Figure 1C and G*), piercing of the membrane and insertion of the stylet into the artificial meal (*Figure 1D–F*), and feeding to full engorgement, as evidenced by the expanded abdomen (*Figure 1E*). *Videos 1*, *2*, *3* and *4* show blood feeding *Ae. albopictus*, *Ae. aegypti*, *An. stephensi*, and *An. coluzzii*, respectively. Imaging the stylet (*Videos 1* and *5*) as it evaluates the artificial meal reveals the striking dexterity of the organ as it rapidly bends, extends, and retracts—aspects of feeding that normally remain hidden inside the skin.

## Automatic characterization of the blood-feeding behavior of multiple species

We created a computational pipeline to extract behavioral statistics from image sequences (see *Figure 1—figure supplement 2* for an overview and Materials and methods for details). The position of individual mosquitoes is tracked over time to yield locomotion statistics (see *Figure 1G* and *Video 6*), and select all time slices that make up a single behavioral trajectory (e.g. landing, exploration, feeding, and take off). The error rate of tracking was 0.045 (5 errors in a validation data set of $n = 111$ tracks, see Materials and methods for details) with the majority of errors arising

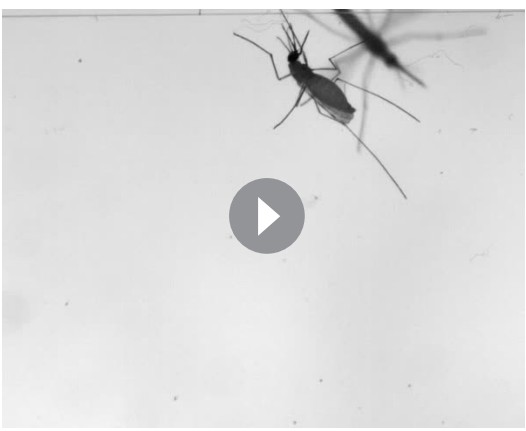

**Video 3.** Several *An. stephensi* females explore the bite substrate, two feed to repletion. The individual that initiates feeding in the top right corner of the frame stops engorging half way, and subsequently moves to the left side of the frame to continue engorging. Video playing in real time.
https://elifesciences.org/articles/56829#video3

from erroneously assigned identities when two mosquitoes cross. Validation videos (see *Video 7* for an example) make it straightforward to manually correct such errors yielding near-perfect tracking. To determine a mosquito's engorgement status, we take advantage of the dilation of the mosquito abdomen when it takes a blood meal (*Figure 1E*). We determine a mosquito's body shape (excluding appendages) using an active contour model to quantify feeding dynamics and engorgement status at each timepoint of a trajectory, and detect full engorgement with a sensitivity of 81% and a specificity of 100% (see *Figure 1* G1-3, *Video 8*, and Materials and methods for details). Together with locomotion statistics, engorgement data provides a high-level description of the behavioral trajectory.

To assess the capability of the biteOscope to characterize the behavior of different species of mosquito, we performed experiments with the two most important vectors of arboviral diseases (*Ae. aegypti* and *Ae. albopictus*) and two dominant malaria vectors (*An. stephensi* and *An. coluzzii*, formerly known as *Anopheles gambiae* M molecular form). *Figure 2* and *Figure 2—figure supplement 1* show locomotion and feeding statistics for the four species. All species land readily on the bite substrate and undertake exploratory bouts leading to full engorgement in 18%, 7%, 4%, and 14% of all trajectories and 46%, 22%, 10%, and 31% of all >10 second trajectories, for *Ae. aegypti*, *Ae. albopictus*, *An. stephensi*, and *An. coluzzii*, respectively, when offered a meal consisting of 1 mM ATP in phosphate buffered saline (PBS). *Figure 2A–D* shows summary statistics of 349 behavioral trajectories of *An. coluzzii* obtained from a total of 1 hr and 15 min of imaging data (five 15-min experiments with 15 females per experiment), demonstrating the throughput of the biteOscope.

*Figure 2E* shows the time spent on the surface versus the distance covered for trajectories that did (large opaque circles) and did not (small transparent dots) lead to full engorgement for the four species. As expected, rather short trajectories do not lead to engorgement, yet less intuitive is the observation that exploratory trajectories that do not lead to engorgement rarely exceed the duration of successful feeding trajectories (8% of non-feeding trajectories takes longer than the mean time to engorge). This suggests that a mosquito's search for blood has a characteristic timescale that is independent of success, and when blood is not found within the time a typical meal takes, the search is aborted.

We further explored this observation using individual *Ae. albopictus* which were offered a bite substrate with a meal of PBS with or without ATP. As PBS alone does not lead to engorgement, mosquitoes offered the PBS only feeder never engorged whereas mosquitoes interacting with the PBS + ATP feeder engorged to full repletion in the majority of cases (55%). High-resolution trajectory analysis enables us to dissect behavioral patterns that lead to (non-)feeding; a trajectory here is defined as landing, the ensuing behavioral sequence, followed by leaving the bite substrate by walking or flying (see *Videos 9* and *10* for two example trajectories). The velocity of a mosquito's centroid can be used to classify locomotion behaviors (stationary, walking, flight) with high accuracy (89% see *Figure 3—figure supplement 1* and Materials and methods for details). *Figure 3* presents ethograms of *Ae. albopictus* on these two bite substrates, and in agreement with the data in *Figure 2E*, shows that trajectories on feeders without ATP (non-feeding) have an approximately equal maximum duration as trajectories leading to full engorgement on the feeder with ATP. While mosquitoes do not increase the duration of exploratory trajectories when not feeding to repletion, the number of exploratory bouts mosquitoes undertook on the PBS only substrate was significantly higher compared to the PBS + ATP case (Wilcoxon rank-sum test p < 0.05), resulting in a slightly longer total exploration time (*Figure 3C*). This suggests that mosquitoes not finding their desired resource increase the frequency with which they initiate searches rather than the duration of individual searches. This observation may be interpreted in the context of the dangers associated with blood-feeding: while on a host, a mosquito runs the risk of being noticed and subsequently killed. When not finding blood, it may therefore be beneficial to abort the search and evacuate from a risky, yet unproductive situation to try elsewhere. The trade-off between exploiting a potential resource versus exploring other options has been shown to depend on the internal state of individuals in other insects (*Katz and Naug, 2015*; *Corrales-Carvajal et al., 2016*), it is possible that such mechanisms play a role here too. *Figure 3* furthermore shows a strong behavioral heterogeneity between individual mosquitoes. While all individuals are from the same mosquito population (and raised and maintained under identical conditions) and interact with the same bite substrate, there is a clear heterogeneity in the number of times a mosquito visits the surface (*Figure 3C*, middle panel), the amount of time she spends exploring the surface (*Figure 3C*, left panel), and the behaviors they

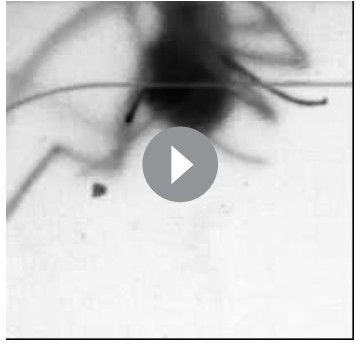

**Video 5.** The stylet of an *Ae. aegypti* female evaluates the artificial meal it finds after piercing the membrane. The stylet is a flexible organ that bends, extends, and retracts in the liquid. Video playing in real time.
https://elifesciences.org/articles/56829#video5

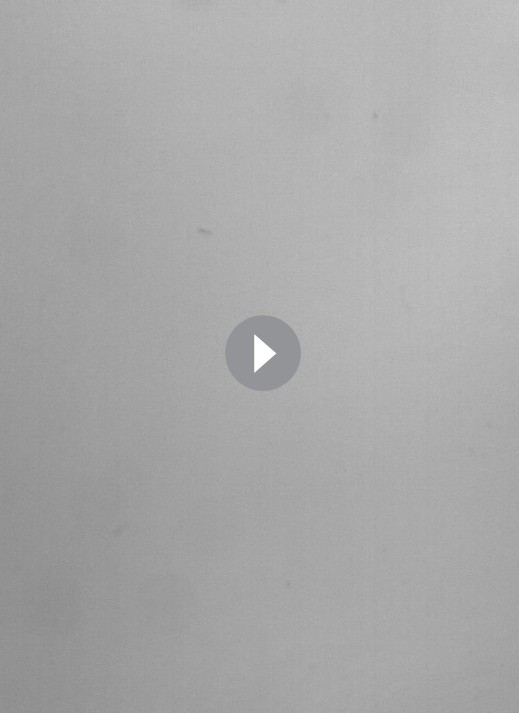

**Video 4.** Several *An. coluzzii* explore interact with the bite substrate, two feed to repletion. Both *Anopheles* species quickly concentrate the obtained meal by excreting liquid (visible as a growing excretion droplet), *Aedes* excrete small droplets as well, yet to a much smaller extent. Video playing in real time.
https://elifesciences.org/articles/56829#video4

engage in. Automatic classification of locomotion behaviors, shows that some individuals often land on the surface to engage in short interactions, while other individuals undertake much longer trajectories. These long trajectories, in turn, vary in the amount of stationary versus locomotion behaviors. The richness of these data highlight the potential of the biteOscope to quantitatively characterize the intricacy of individual behaviors hidden in population averages.

## Pose estimation, behavioral classification, and contact-dependent sensing

We next turned to body part tracking to acquire a more detailed description of behavioral trajectories. Body part tracking is powerful to address a variety of questions, for example by determining points of surface contact of specific appendages, or to estimate the pose of an animal, which when tracked over time can be translated into a behavioral sequence. We used a recently developed deep learning framework, DeepLabCut (*Mathis et al., 2018*), to train a convolutional neural network (CNN) to detect the head, proboscis, abdomen, abdominal tip, and six legs of *Ae. aegypti* and *Ae. albopictus*. Due to their morphological similarity, the same CNN can be used to track the body parts of both *Aedes* species with a mean accuracy of 11 pixels (275 micrometer, see Materials and methods for details) in a 4.3 × 4.3 cm field of view. Tracking stylet insertions into the artificial meal during probing and feeding using DeepLabCut was challenging, and therefore not included.

*Figure 4A–C* shows body part tracking results of *Ae. albopictus* and reveals the choreography of three distinct behaviors. Anterior grooming is characterized by circular motion of the forelegs followed by the proboscis, while the middle legs remain stationary (see *Figure 4—video 1*). During walking, the tips of all six legs oscillate along the body axis while the proboscis explores laterally (see *Figure 4—video 2*), while during probing, the fore and middle legs pull toward the body and the proboscis remains stationary (see *Figure 4—video 3*). Inference is done on raw images and the obtained coordinates thus subject to movement of the mosquito. To correct for this, the coordinates are translated and rotated to align along the body axis taking the abdominal tip as the origin. *Figure 4D–I* shows time series of the obtained egocentric coordinates and their corresponding wavelet transforms. The three behaviors each are associated with distinct periodic movements: smooth periodic motion of the forelegs during anterior grooming (x, and y coordinates), punctuated oscillations along the body axis during walking (x coordinate), and faster jerky movement during probing (x, and y coordinate of forelegs, y coordinate of middle legs). These trajectories can be

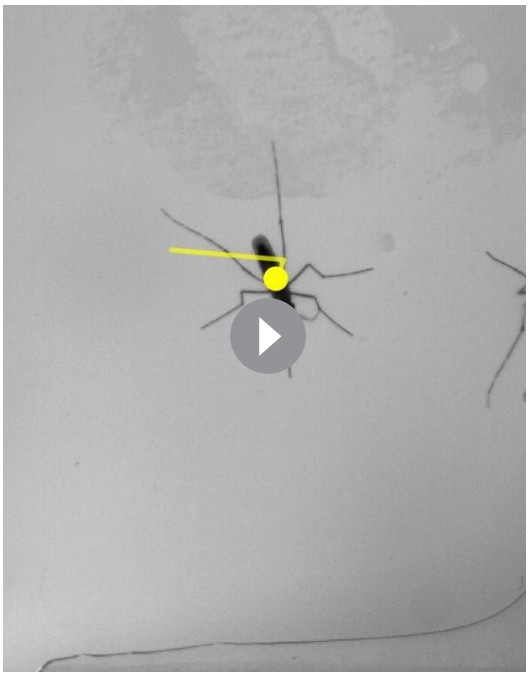

**Video 6.** Tracking the centroid of *Ae. aegypti*. The color of the centroid and the trail is a measure for the instantaneous velocity of the animal.

https://elifesciences.org/articles/56829#video6

used in concert with locomotion and body-shape features as inputs for behavioral classification algorithms. The data outputted by our computational pipeline is ideally suited for classification in either a supervised (e.g. *Kain et al., 2013*; *Kabra et al., 2013*) or unsupervised (e.g. *Berman et al., 2014*; *Marques et al., 2018*; *Calhoun et al., 2019*; *Tao et al., 2019*) approaches (see *Figure 4—figure supplement 1*).

## DEET repels *An. coluzzii* upon contact with legs

Next, we explored the use of body part tracking within the context of contact-dependent sensing by *An. coluzzii*. *Anopheles* and *Aedes* mosquitoes have an overall similar body plan, yet the length of their maxillary palps (an olfactory appendage projecting from the head) is very different with anophelines having maxillary palps with a length comparable to the proboscis, while *Aedes* palps are much shorter. We therefore trained a CNN for *Anopheles* body parts, which additionally tracks the position of the maxillary palps (mean accuracy for *Anopheles* body parts: six pixels, 150 micrometer). Through this approach, we addressed the open question if *An. coluzzii* is repelled by N,N-diethyl-meta-toluamide (DEET) upon contact. DEET has been in use as an effective insect repellent for decades

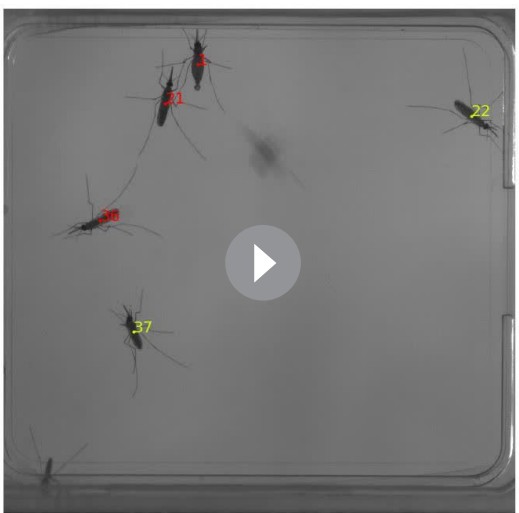

**Video 7.** Centroid tracking of *An. coluzzii*. Example of a validation video for tracking data playing at 2.5 times reduced speed. Numerical IDs are assigned to mosquitoes and shown overlaid on the original data (the position of the centroid is indicated in the same color as the ID).

https://elifesciences.org/articles/56829#video7

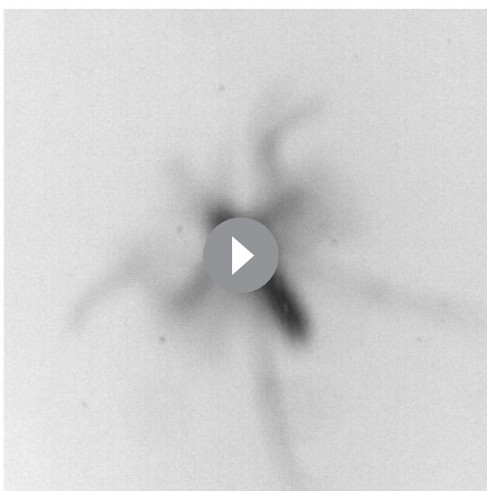

**Video 8.** The abdomen of an *Ae. aegypti* female expands dramatically during blood feeding. Fitting an active contour model to the mosquito body (after computationally removing appendages) provides the abdomen width (and other shape parameters) which can be used to estimate engorgement status.

https://elifesciences.org/articles/56829#video8

and is thought to act on mosquitoes through several mechanisms that are either olfactory- or contact-based (*DeGennaro, 2015*). *Afify et al., 2019* recently observed that volatile DEET does not activate olfactory neurons in *An. coluzzii* and reported that *An. coluzzii* does not avoid DEET by smelling it (*Afify et al., 2019*; *Afify and Potter, 2020*). *Afify et al., 2019* proposed that DEET may prevent *An. coluzzii* from locating humans by masking odorants emanating from potential hosts. However, it remained an open question if *An. coluzzii* is repelled by DEET upon direct contact.

We addressed this question by imaging *An. coluzzii* offered a bite substrate partly coated with DEET. *Figure 5* shows that *An. coluzzii* do land on both the DEET-coated and uncoated surface, and there is a moderate decrease in landing rate on the DEET-coated portion (the landing rate is 1.9 times lower, normalized for surface area). The time *An. coluzzii* spend on the DEET-coated surface, however, is much shorter: trajectories on the DEET-coated surface ($n = 34$) are on average seven times shorter when compared to the uncoated surface ($n = 412$). Furthermore, the longest residence time observed on the DEET-coated surface was less than 6 s, whereas individual *An. coluzzii* spent up to 52 s on the uncoated surface. From these data, we conclude that *An. coluzzii* do approach and land on the DEET-coated surface, but avoid (prolonged) contact with it, indicating that *An. coluzzii* indeed is not strongly repelled by volatile DEET at very close range, yet avoids it on contact.

We next asked what appendages mediate this contact dependent avoidance. The 34 trajectories in which *An. coluzzii* visited the DEET area consisted of 25 'touch and go' events in which an individual approached the DEET surface in flight, landed, and immediately took off after first contact (residence time on DEET surface <0.5 second, see *Video 11* for a typical 'touch and go' event played at 1/4 speed). In the remaining nine trajectories, *An. coluzzii* landed outside the DEET area and moved onto it (see *Figure 5* and *Video 12*), the reverse scenario in which a mosquito would land on the DEET surface and move onto the non-coated surface was never observed. We performed body part tracking on the trajectories where *An. coluzzii* moved from the non-coated surface to the DEET-coated surface and developed analysis software that scores how often a specific body part visits an

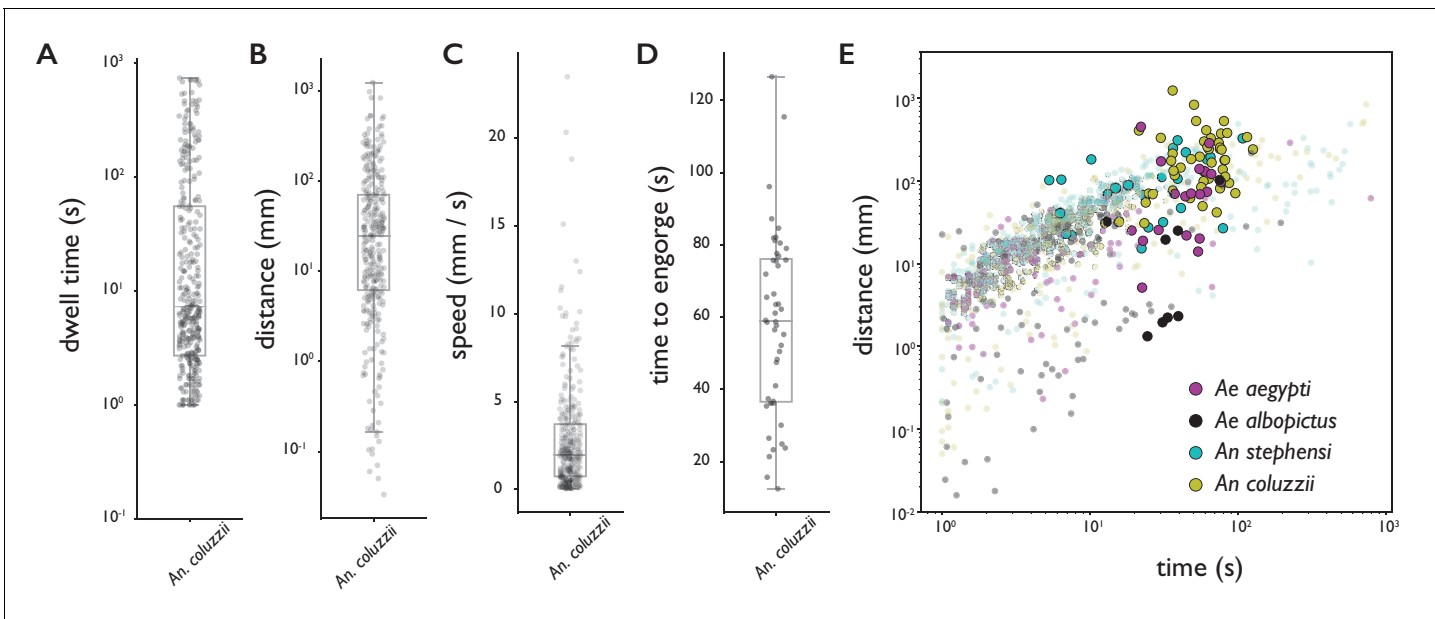

**Figure 2.** Behavioral statistics of *An. coluzzii* (A–D) and all four species (E). Each datapoint is derived from an individual trajectory, boxes indicate quartiles. (A) The time spent on the bite surface ($n = 349$). (B) The total distance covered walking on the surface during a trajectory ($n = 349$). (C) The mean velocity during a trajectory ($n = 349$). (D) The time from landing to full engorgement (for trajectories leading to full engorgement, $n = 48$). (E) The duration of a trajectory (total time for trajectories not leading to engorgement (transparent dots), time to full engorgement for trajectories that led to full engorgement (opaque circles)) versus the distance covered during that trajectory. The different colors denote different species, *Ae. aegypti*: magenta, *Ae. albopictus*: black, *An. stephensi*: cyan, *An. coluzzii*: yellow.

The online version of this article includes the following source data and figure supplement(s) for figure 2:

**Source data 1.** Source data for all four species provided as Pandas DataFrames.
**Figure supplement 1.** Behavioral statistics of *Ae. aegypti*, *Ae. albopictus*, *An. stephensi*, and *An. coluzzii*.

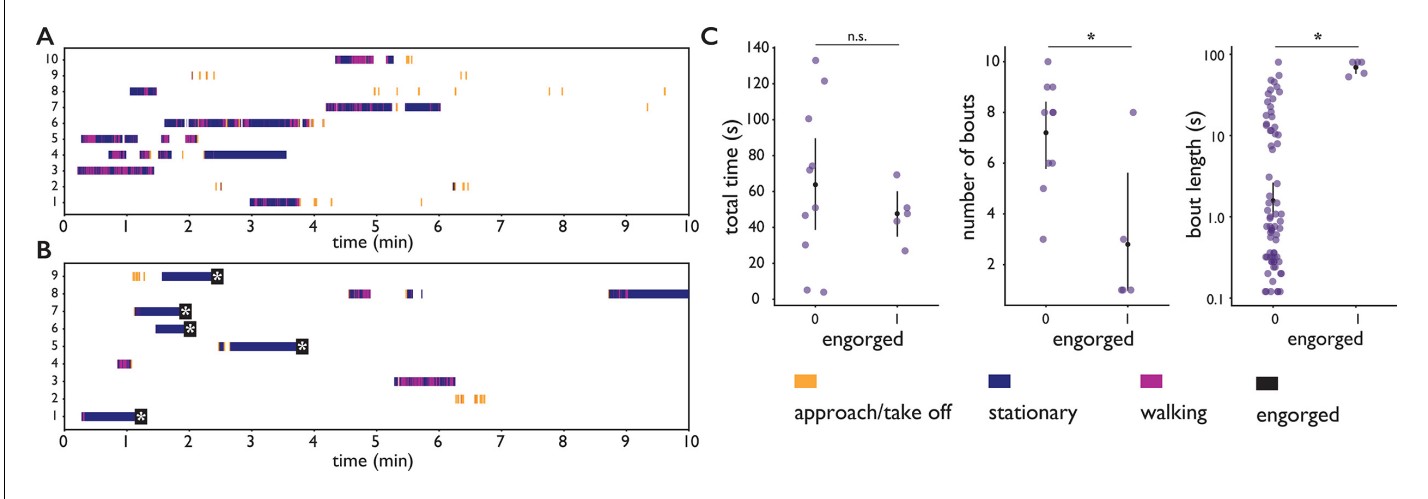

**Figure 3.** Feeding behavior of individual *Ae. albopictus*. (A, B) Ethograms of individual *Ae. albopictus* interacting with a bite substrate offering a PBS only meal, $n = 10$ (A), and a meal consisting of PBS + 1 mM ATP, $n = 9$ (B). Distinct exploratory bouts appear as continuous blocks in the ethogram and are labeled according to the behavior being displayed: flight (yellow), walking (purple), and stationary (dark blue), the time of engorgement to full repletion is marked by a black box and a white asterix. (C) Behavioral statistics of the data displayed in A and B showing the total time spent on the bite substrate (left, no significant difference $p = 0.39$, Wilcoxon rank-sum test), the number of exploratory bouts undertaken (middle, significantly different $p = 0.020$, Wilcoxon rank-sum test), and the length of individual bouts (right, significantly different $p = 9 \times 10^{-4}$, Wilcoxon rank-sum test), of *Ae. albopictus* exploring the PBS only substrate (labeled 0) and those that engorged to full repletion on the PBS + ATP substrate (labeled 1). Individual data points are shown in purple, the mean and associated 95% confidence interval are depicted by a black dot and bar, respectively. Individuals that were offered the PBS + ATP substrate but did not feed to full repletion were excluded from this analysis.

The online version of this article includes the following source data and figure supplement(s) for figure 3:

**Source data 1.** Source data for all experiments with individual *Ae. albopictus* females.
**Figure supplement 1.** The accuracy of automatic classification of locomotion behaviors (stationary, walking, flight) is 89% and exceeds 80% for a range of parameter values.

arbitrarily shaped region of interest. We observed that the legs of individuals came in contact with the DEET surface in all cases, whereas the proboscis only came in contact with the DEET surface in 5/9 cases (in cases where no proboscis contact was observed, the entire proboscis remained outside the boarders of the DEET-treated area). Together, these observations demonstrate that *An. coluzzii* are indeed repelled upon contact with DEET, and indicate that this behavior is mediated by sensilla on the legs, and likely not the proboscis. While contact-dependent sensing (e.g. by tarsal neurons) seems the most plausible mechanism to explain this contact-dependent avoidance, we cannot rule out that physical properties of the DEET coating play a role as well.

## Discussion

The biteOscope provides an alternative for current methods using human subjects or mice to study mosquito blood feeding. The elimination of the need for a human subject opens new avenues of research, for example allowing blood-feeding studies with pathogen-infected mosquitoes, enabling precise surface manipulations and characterization of the associated behavior, and facilitates the use of high-resolution imaging and machine-learning-based image analysis. Through these innovations, the biteOscope increases experimental throughput and expands the type of experiments that can be performed and measurements that can be made. We developed computational tools that allow the behavioral monitoring of mosquitoes at an unprecedented level of detail. Behavioral research on other animals, including fruit flies (*Werkhoven et al., 2019*; *Pereira et al., 2019*) and zebrafish (*Marques et al., 2018*; *Johnson et al., 2020*) shows that high spatiotemporal resolution data describing the posture of animals can be very informative to dissect behavioral trajectories and compare behavioral statistics across individuals and experimental treatments. While the details of computational approaches differ, a common theme is the two dimensional embedding of a high-dimensional representation of an animal at a given time point (e.g. body part coordinates and

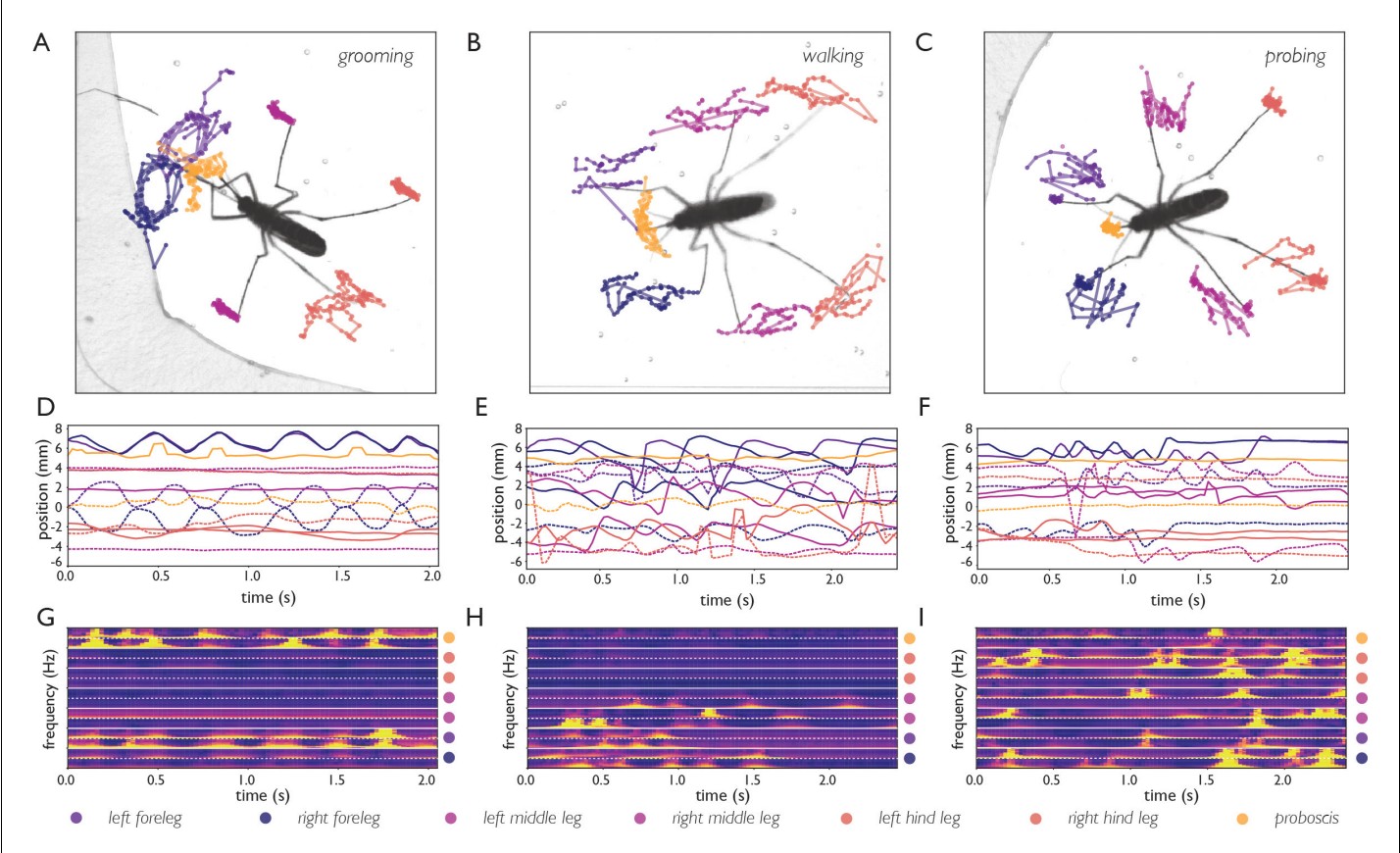

**Figure 4.** Body part tracking reveals movement patterns of specific behaviors. Color coding of plots in panels A-F are displayed at the bottom of the figure. (A–C) Trajectories of the tips of the six legs and proboscis of an *Ae. albopictus* female grooming her antennae (A), walking (B), and probing (C). (D–F) Time traces showing egocentric x (full lines) and y (dashed lines) coordinates of the body parts of mosquitoes shown in A-C. Anterior grooming is characterized by smooth periodic movement in the x and y planes. During walking the x-coordinate shows a swing that alternates between fore, middle, and hind leg; probing shows rapid pulling of the fore and middle legs towards the body. (G–I) Continuous wavelet transforms of the body part coordinates highlight the periodicity of movements. The amplitude of the spectrogram is indicated by the color, going from low (purple) to high (yellow). Yellow bands indicate periodic movement of a body part. Spectrograms of the seven body parts are stacked and separated by white lines (color coding on the right shows stacking order, with the x-coordinate of the body part on top, and y-coordinate on the bottom (x, and y coordinates are separated by a dashed line)).

The online version of this article includes the following video and figure supplement(s) for figure 4:

**Figure supplement 1.** Two-dimensional embedding of data shown in A-I.

**Figure 4—video 1.** Body part tracking of *Ae. albopictus* anterior grooming, corresponding to *Figure 4A,D,G*.
https://elifesciences.org/articles/56829#fig4video1

**Figure 4—video 2.** Body part tracking of *Ae. albopictus* walking, corresponding to *Figure 4B,E,H*.
https://elifesciences.org/articles/56829#fig4video2

**Figure 4—video 3.** Body part tracking of *Ae. albopictus* probing, corresponding to *Figure 4C,F,I*.
https://elifesciences.org/articles/56829#fig4video3

derived features), data points in two dimensions can subsequently be clustered to reveal behavioral classes (see *Figure 4—figure supplement 1* for an illustration of this concept using tSNE to embed the data represented in *Figure 4*). Translating such advances in computational ethology to mosquito research is a very promising avenue for future research.

We used the biteOscope to describe behavioral patterns of four medically relevant mosquito species and anticipate that such datasets will provide a useful 'behavioral baseline' for future studies quantifying the effect of a mosquito's physiology on blood feeding behavior. The role of pathogen infections is particularly interesting in this respect, as infections may alter feeding behavior, for example by affecting the structural integrity of the salivary glands or other tissues, or inducing

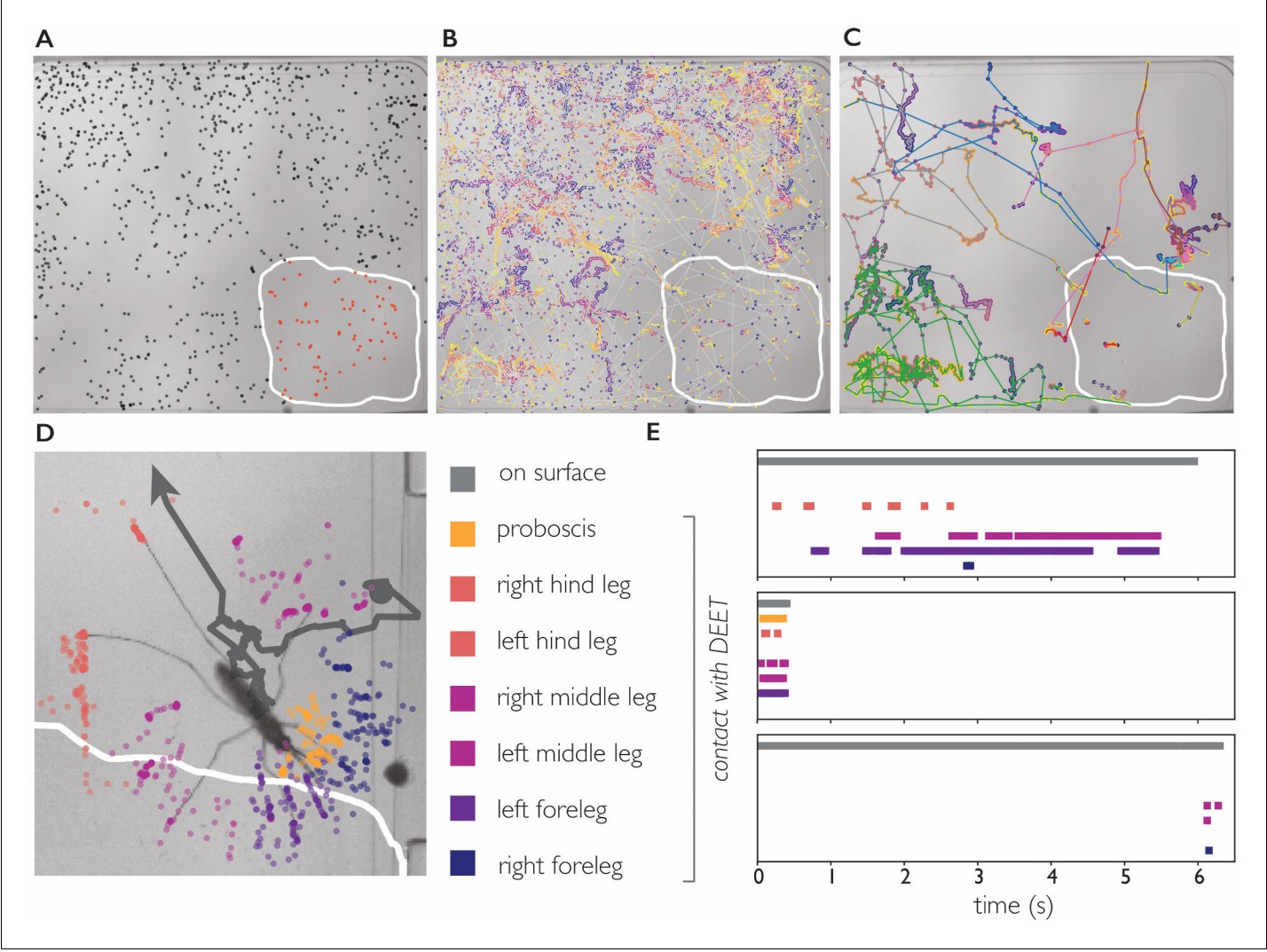

**Figure 5.** DEET repels *An. coluzzii* on contact with legs. (A) Landings on a substrate partly coated with 50% DEET (white line indicates DEET-coated surface). Black dots indicate landings outside the DEET area, red dots indicate landings inside the DEET area. The landing rate in the DEET area is approximately 1.9 times lower compared to the non-treated surface. (B) Trajectories of mosquito movement on the surface. Dots of individual tracks are colored from purple (start of the track) to yellow (end of the track). *An. coluzzii* on average spend seven times longer on the non-coated surface compared to the DEET-coated surface. (C) Example tracks of mosquitoes landing on the non-treated area and subsequently entering the DEET-coated area. (D) Body part tracking of a mosquito near the edge of the DEET-coated surface. The grey line indicates the movement of the center of mass of the mosquito (a dot indicates the start of the track, arrowhead departure). Colored dots indicate the position of the legs and proboscis during the section of the trajectory where the mosquito is within reach of the DEET-coated area (indicated by the white line). (E) Ethogram showing typical behavioral patterns when a mosquito comes in contact with DEET. The grey bar (top) indicates that a mosquito is anywhere on the surface (including the uncoated area), the colored bars indicate contact of a specific appendage with DEET. The top panel corresponds to the mosquito shown in (D) illustrating a mosquito that walks toward the DEET area, contacts it with several legs, and flies away. The middle panel is an example of 'touch and go' contact in which a mosquito lands on the DEET area, contacts it with several legs and proboscis, and takes off. The bottom panel shows a mosquito that after a long exploratory bout outside the DEET area, takes off as soon as the right foreleg and both middle legs contact the DEET area.

systemic change through the immune system or infection of neural tissues (*Rossignol et al., 1984*; *Girard et al., 2007*; *Cator et al., 2013*; *Turley et al., 2009*). A quantitative understanding of such behavioral alterations, however, is lacking. Gaining such insights is of high epidemiological relevance, as mathematical models suggest that (pathogen induced) changes in bite behavior can have important implications for pathogen transmission (*Cator et al., 2014*; *Abboubakar et al., 2016*). In addition to pathogen-induced behavioral changes, there are many other promising lines of inquiry, including the behavioral influence of the microbiome (*Dickson et al., 2017*), which, in other insects

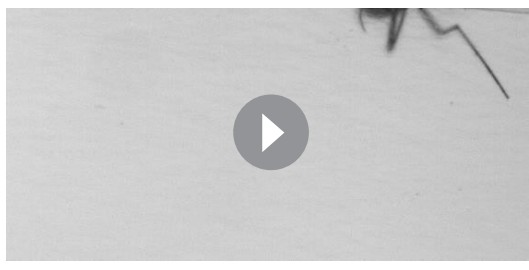

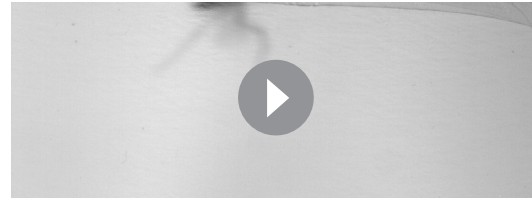

**Video 9.** *Ae. albopictus* female walking onto the bite substrate (artificial meal of PBS without ATP), probing the substrate several times, and moving away. Video playing in real time.
https://elifesciences.org/articles/56829#video9

**Video 10.** *Ae. albopictus* female exploring the surface of PBS only feeder (without ATP). While walking, the proboscis often moves laterally and taps the surface. Video playing in real time.
https://elifesciences.org/articles/56829#video10

such as *Drosophila*, influences locomotor behavior (*Schretter et al., 2018*) and food choice (*Leitão-Gonçalves et al., 2017*; *Wong et al., 2017*). *Drosophila* research furthermore shows interesting examples of collective behaviors mediated by for example olfaction or direct contact between animals (*Schneider et al., 2012*; *Ramdya et al., 2015*; *Lihoreau et al., 2016*; *Ramdya et al., 2017*), it would be interesting to explore if mosquitoes also take advantage of collective intelligence when searching for food or avoiding noxious stimuli. Tools enabling high-throughput behavioral monitoring may also be useful to characterize population intervention strategies aimed at curbing pathogen transmission, such as *Wolbachia* infected *Ae. aegypti*, or *Anopheles* genetically engineered to be refractory to *P. falciparum* infection. Quantifying the behavioral effects of such interventions is an important step toward assessing the competitiveness of engineered mosquitoes in the field. As the biteOscope enables novel high-throughput experiments with a variety of mosquito species, we anticipate that it will prove useful for the characterization of various behaviors relevant to pathogen transmission.

By tracking the individual body parts of *An. coluzzii*, we discovered that they are repelled by DEET upon leg contact—a mechanism that may work in concert with other ways in which DEET pre-

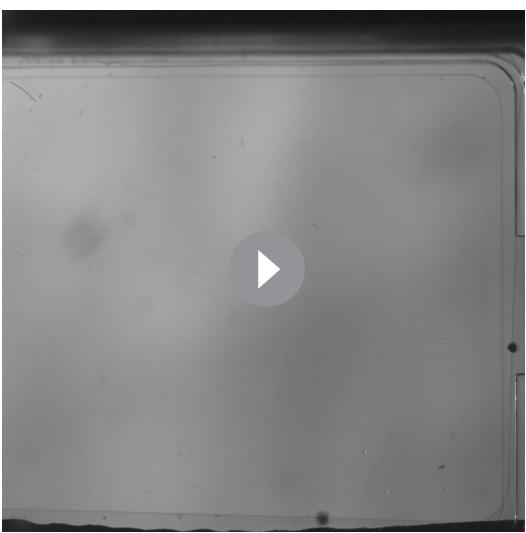

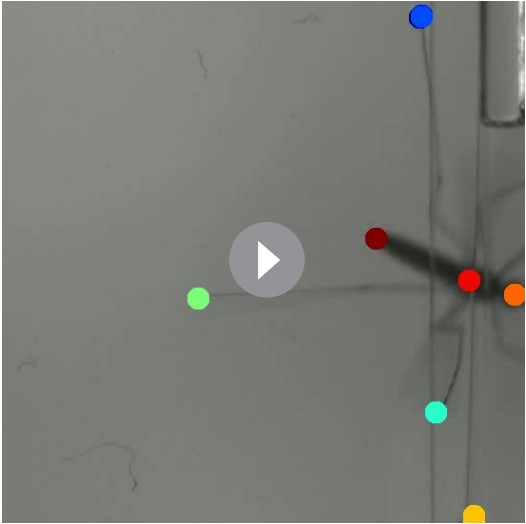

**Video 11.** *An coluzzii* landing on the DEET-coated surface and immediately taking off. Video playing four times slower than real time. The majority of trajectories in which *An. coluzzii* comes into contact with the DEET-coated surface results in an immediate take off.
https://elifesciences.org/articles/56829#video11

**Video 12.** *An. coluzzii* moving onto the DEET-coated surface. Body part tracking shows that this female lands outside the DEET-coated area and subsequently her left fore and middle leg come into contact with the DEET-coated portion. After a short contact, the mosquito flies away.
https://elifesciences.org/articles/56829#video12

vents anopheline mosquitoes to locate humans. Our findings regarding *An. coluzzii* are in agreement with observations in *Ae. aegypti* which are also repelled by DEET upon leg contact (*DeGennaro et al., 2013*; *Dennis et al., 2019*). However, in contrast to *An. coluzzii*, olfactory neurons of *Ae. aegypti* are activated by volatile DEET (*Davis and Rebert, 1972*; *Boeckh et al., 1996*; *Stanczyk et al., 2010*) and *Ae. aegypti* has been reported to avoid volatile DEET in recent studies (*Stanczyk et al., 2013*; *Afify and Potter, 2020*) (in contrast, an earlier study reported attraction of *Ae. aegypti* by DEET [*Dogan et al., 1999*]). Together, these observations suggest that contact-based repellency may be conserved across *Anopheles* and *Aedes* mosquitoes and thus may be a potentially interesting target for the design of new repellents. It is less clear, however, what degree of conservation exists for the olfactory modes of action, as the only study comparing the olfactory effects of volatile DEET on *Anopheles* and *Aedes* mosquitoes in the same assay, suggests that the former is not repelled at all by volatile DEET, while the latter showed moderate repulsion (these behavioral responses may be concentration dependent) (*Afify and Potter, 2020*). This observation, together with the observation that volatile DEET activates olfactory neurons in *Ae. aegypti* while it does not seem to do this in *An. coluzzii*, suggest that volatile DEET may modulate the response of olfactory neurons to attractive stimuli ('scrambling of the odor code' [*Pellegrino et al., 2011*]) and/or trigger repulsion in *Ae. aegypti*, while these mechanisms seem less appropriate for *An. coluzzii*. In addition to effects on olfactory signaling, DEET has also been suggested to decrease the amount of volatile odorants emanating from hosts through chemical interactions between DEET and the odorants resulting in the masking of a host (*Afify et al., 2019*). As in this scenario the amount of attractive odorants reaching a mosquito is reduced, it may affect the behavior of a variety of species. The observation that both *Ae. aegypti* and *An. coluzzii* avoid DEET upon leg contact, while the effects of volatile DEET may partly overlap and partly differ, may guide efforts aimed at uncovering the underlying molecular mechanisms.

Our results highlight the use of body part tracking in assigning roles to the various sensory appendages the mosquito body has. The recent surge in genetic tools available to manipulate mosquitoes is shedding light on the genetic elements that mediate pathogen transmission relevant behaviors (*Matthews et al., 2019*; *Ingham et al., 2020*; *Raji et al., 2019*; *Greppi et al., 2020*). Combining such molecular level insights with detailed behavioral tracking and chemical surface patterning, may enable a deep understanding of how contact-dependent sensing drives blood feeding, and other important phenotypes such as insecticide resistance and egg laying preferences.

When studying animal behavior in the lab a trade-off exists between the level of experimental control and detail of observation on the one hand, and an accurate representation of natural conditions and behaviors on the other. In case of the biteOscope, an engineered bite substrate opens up a variety of possibilities including surface modifications and high-resolution imaging impossible on human skin, yet the bite substrate does not offer the full set of cues (and thus behavioral responses) a human host would. It would therefore be interesting to add more human-associated cues, for instance using materials that resemble the texture of skin, or by coating the bite substrate with attractive human odorants (*Okumu et al., 2010*). In addition to more closely mimicking human hosts by presenting olfactory stimuli, surface coatings could be used to dissect the role of contact-dependent gustatory behaviors on the skin surface in bite site selection. It is important to note that many of the factors that may change behavior mentioned above (e.g. infections/nutritional status or components of the microbiome) are best assessed in a relative manner, for example comparing non-infected to infected individuals. Comparing cohorts of mosquitoes undergoing different experimental treatments puts less emphasis on the absolute attractiveness of the bite substrate and thus mitigates potential issues related to the fact that a synthetic bite substrate is likely less attractive than a real live host.

We took advantage of the possibility to elicit engorgement on a transparent meal to facilitate imaging. It seems feasible to add a dye to the meal to provide visual cues to the mosquito without interfering with image quality. Using whole blood, however, is challenging in the current system. It would therefore be worthwhile to explore the use of microfluidics to incorporate blood flow into the bite substrate while maintaining optical access. A recent study took advantage of the biteOscope to quantify stylet contact with artificial meals (*Jove et al., 2020*), combining such efforts with artificial vasculature presents exciting opportunities to characterize the role of the stylets in the search for blood.

The biteOscope is designed with a variety of possible users in mind. It has a relatively modest price tag (900–3500 USD depending on the configuration), uses readily available materials and components, and when disassembled fits in a backpack—characteristics we hope will facilitate adoption. Beyond the lab, we foresee interesting applications of the behavioral tracking of mosquitoes in (semi-)field settings, and expect that innovative tools that provide high-quality quantitative data will enable discoveries in this space. We anticipate that the techniques and computational tools presented here will provide a fresh perspective on mosquito behaviors that are relevant to pathogen transmission, and enable researchers to gain a detailed understanding of blood feeding without having to sacrifice their own skin.

# Materials and methods

**Key resources table**

| Reagent type (species) or resource | Designation | Source or reference | Identifiers | Additional information |
|---|---|---|---|---|
| Strain, strain background (*Ae. aegypti*) | KPPTN | Lambrechts lab, Institut Pasteur | | Thailand, G18 |
| Strain, strain background (*Ae. aegypti*) | D2S3 | BEI resources | | Puerto Rico x Nigeria cross |
| Strain, strain background (*Ae. aegypti*) | Liverpool | Vosshall lab, Rockefeller University | | West Africa |
| Strain, strain background (*Ae. albopictus*) | BP | Lambrechts lab, Institut Pasteur | | Vietnam, G23 |
| Strain, strain background (*An. stephensi*) | Sda500 | CEPIA, Institut Pasteur, Paris | | Pakistan |
| Strain, strain background (*An. coluzzii*) | N'Gousso | CEPIA, Institut Pasteur, Paris | | Cameroon |
| Software, algorithm | biteOscope code | this paper | | *Hol, 2020* github.com/felixhol/biteOscope (copy archived at https://github.com/elifesciences-publications/biteOscope). |
| Software, algorithm | DeepLabCut | *Mathis et al., 2018* | | version 2.0.9 |

## Mosquito rearing and maintenance

The mosquito species/strains used in this study are described in Key resources table. Larvae were hatched and reared in water at a density of approximately 200 larvae per liter on a diet of fish food. Adult mosquitoes were maintained at 28, 75% relative humidity, and a photoperiod of 12 hr light : 12 hr dark in 30 × 30 × 30 cm screened cages having continuous access to 10% sucrose. Prior to experiments, mosquitoes were deprived of sucrose for 6–12 hr while having access to water. Mosquitoes aged 6–25 days old were used for behavioral experiments. Experiments using *Ae. aegypti* and *Ae. albopictus* were performed during light hours, while experiments with *An. stephensi* and *An. coluzzii* were performed during dark hours. Mosquitoes had no access to water during experiments.

## biteOscope hardware

A full list of components necessary to build the biteOscope is available in *Appendix 1—table 1*. Depending on the experimental requirements, several components can be easily adapted (e.g. cage geometry or bite substrate) or replaced by more economical alternatives (e.g. imaging components).

### Cage, bite substrate, and environmental control

Cages were constructed from 1.59 mm (1/16 inch) thick clear cast acrylic sheets (McMaster Carr) cut to the required dimensions using a laser cutter (Epilog). To facilitate mounting of the bite substrate,

an opening having the same dimensions as the bite substrate was cut in the floor or one of the walls of the cage (all design files are available on Github). We noted that orientation of the bite substrate affects both the landing rate of mosquitoes (e.g. *Ae. albopictus* had a lower landing rate on vertically mounted substrates compared to those mounted in the floor) and their orientation (on vertical surfaces mosquitoes aligned with gravity, head up bottom down). While this suggests that orientation is an interesting parameter to explore, all experiments presented here were performed with floor-mounted substrates to prevent behavioral biases possibly associated with vertically mounted substrates. The bite substrate was made using a 70-mL culture flask (Falcon 353109) filled with warm water maintained at 37 by a Raspberry Pi taking the input of a waterproof temperature probe (DS18b20, Adafruit) to control a Peltier element (digikey) used for heating. If desired, the same Raspberry Pi can operate a 12 Volt solenoid valve (Adafruit) to control the inflow of gas. An artificial meal of phosphate buffered saline (sigma-aldrich) (supplemented with 1 mM of adenosine triphosphate (sigma aldrich) where noted) was applied to the rectangular section of the outside of the culture flask and covered with a Parafilm membrane. This creates a fluid cell supported by the membrane and the outside of the culture flask. The artificial meal is maintained at 37 by the water inside the flask. We performed additional experiments using 1 mM ATP in 110 mM NaCl and 20 mM $NaHCO_3$ as the artificial meal and observed robust feeding of *Aedes* and *Anopheles* mosquitoes on this formulation as well.

## Imaging and illumination

Images were acquired at 25 or 40 frames per second using a Basler acA2040-90um camera controlled using Pylon 5 software running on an Ubuntu 18.04 computer (NUC8i7BEH). The camera was equipped with a 100 mm macro lens (Canon macro EF 100 mm f/2.8L). Illumination for *Aedes* experiments was provided by two white light LED arrays (Vidpro LED-312), while IR LEDs (Taobao) were used for *Anopheles* experiments. The same camera was used for white light and IR illuminated experiments. Thorlabs components were used to arrange all optical components and the experimental cage at suitable distance.

## Computational tools

All image processing and downstream analysis code was written in Python 3 and is available from Github (https://github.com/felixhol). Raw images were background subtracted, thresholded, and subjected to a series of morphological operations to yield binary images representing mosquito bodies of which the center of mass was determined using SciPy (*Virtanen et al., 2019*). The Crocker–Grier algorithm (*Crocker and Grier, 1996*) was used to link the obtained coordinates belonging to an individual mosquito in time using trackPy (*Allan et al., 2016*). The obtained tracking data is used to select all images that make up a single behavioral trajectory (e.g. landing, exploration, feeding, and take off) and store cropped image sequences centered on the focal mosquito. In addition to the computationally extracted data described below, such image sequences can also be used for the manual annotation of other events (e.g. stylet insertion as done in *Jove et al., 2020*).

We verified the tracking results of 111 individual trajectories across 12293 images resulting in an error rate of 0.045 (5/111). The validation dataset includes data from both *Aedes* and *Anopheles* experiments and consists of images having a variety of densities ranging from 0.05 to 0.4 mosquitoes per $cm^2$. The most common error (4/5) is caused by wrongly assigning the identity of two mosquitoes that cross (e.g. an individual moving over another one and thus overlapping in the image). Interestingly, the validation videos (e.g. *Video 7*) make it straightforward to correct such errors by manually re-assigning the correct identity to the track. A rather minor amount of manual interventions therefore results in nearly perfect tracking.

## Classifying locomotion behaviors

Locomotion behaviors (as presented in *Figure 3A and B*) can be automatically assigned based on the velocity of the centroid of a mosquito. To estimate the accuracy of this procedure, we manually labeled the behavior *Ae. albopictus* mosquitoes exhibited in 1124 frames of the dataset presented in *Figure 3* and compared the labeled behaviors to the computationally detected behaviors. The overall accuracy of behavioral classification was 89%, with a per class accuracy of 90% (stationary), 89% (walking), and 97% (flight), with accuracy defined as: $\frac{TP+TN}{O}$, with TP denoting true positives, TN

true negatives, and O the number of observations. The classification of locomotion behaviors depends on the velocity thresholds set to distinguish flight, walking, and stationary behaviors. *Figure 3—figure supplement 1* shows that classification accuracy peaks at 89% accurate classifications using a stationary – walking threshold of 2 mm/s and a walking – flight threshold of 12 mm/s, and exceeds 80% for a range of parameters.

## Detecting engorgement

Images cropped on the focal mosquito (above) are used to determine a mosquito's body shape at each timepoint to infer engorgement status by computationally removing all appendages and fitting an active contour model (using OpenCV [*Bradski, 2000*]) to the remaining body shape. For a mosquito to be computationally defined as engorged, two empirically determined conditions need to be met:

1. The abdominal area needs to expand 1.3 fold. Fold expansion is calculated as the ratio of the 90th percentile of abdominal area along the full trajectory and the 10th percentile of abdominal area in the first 10 s of the trajectory.
2. The 90th percentile of abdominal area measurements needs to exceed 2.4 mm$^2$ for *An. stephensi* and *An. coluzzii*, or 3.0 mm$^2$ for *Ae. aegypti* and *Ae. albopictus*.

We estimated the performance of the engorgement detection algorithm by validating all data presented in *Figure 2* and *Figure 2—figure supplement 1* and observed an overall sensitivity of engorgement detection $\frac{TP}{P} = 0.81$ ($n = 130$), with a sensitivity of 0.97 ($n = 29$) and 0.76 ($n = 101$) for *Aedes* and *Anopheles* mosquitoes, respectively. The overall specificity was $\frac{TN}{N} = 1.0$ ($n = 1184$). The difference in sensitivity for detecting engorgement in *Aedes* versus *Anopheles* may have two reasons: (1) *Anopheles* excrete excess liquid during feeding to a much larger extent than *Aedes* mosquitoes, resulting in a less pronounced dilation of the abdomen and (2) some *Anopheles* experiments had a higher density of mosquitoes on the bite substrate leading to mosquitoes touching more often resulting in less accurate fitting of the body shape.

## Body part tracking

The DeepLabCut framework (*Mathis et al., 2018*) was used to train a convolutional neural network (ResNet architecture) to detect the most distal part of the six legs, the abdominal tip, the center of the abdomen, the head, the tip of the proboscis, and for anophelines the tip of the maxillary palps. Due to their similar appearance, *Ae. aegypti* and *Ae. albopictus* can be analyzed using the same network, while a second network was trained for *An. stephensi* and *An. coluzzii*. Approximately 350 images were used to train the *Aedes* dataset, while approximately 300 images were used for the *Anopheles* dataset. To ensure robustness of training, the *Aedes* and *Anopheles* models were trained on 4 and 2 shuffles of the training set, respectively. Averaged across shuffles training yielded an accuracy, defined as the mean average Euclidean error between the manual labels and predicted labels, of 11 pixels (275 μm) and six pixels (150 μm) in a 4.3 × 4.3 cm field of view, for *Aedes* and *Anopheles,* respectively. In addition to the mean performance across all body parts, prediction accuracies per groups of body parts (core: head, proboscis, abdomen, abdominal tip, (and palps for *Anopheles*); and legs: tips of all six legs) was 1.7 pixels (43 μm) for core body parts, and 1.6 pixels (40 μm) for the tips of legs for the best performing *Aedes* model; and 5.2 pixels (130 μm) and 3.7 pixels (93 μm) for core and legs for the best performing *Anopheles* model. Trained models are available on GitHub.

We used cropped image sequences (described above) for inference. To facilitate downstream analysis of body part tracking data, body part coordinates can be aligned along the body axis (defined along the abdominal tip and center of the abdomen) yielding coordinates invariant of body orientation or movement. The wavelet transforms shown in *Figure 4* are obtained by applying the Morlet continues wavelet transform to this data. Two-dimensional embedding of the aligned body part coordinates and their wavelet transform (*Figure 4—figure supplement 1*) was done by scaling the data (subtracting the mean and scaling to unit variance) and using t-distributed stochastic neighborhood embedding (tSNE) in two dimensions (*Maaten and Hinton, 2008*).

## Experiment-specific procedures

### Feeding experiments

Population experiments (*Figure 1* and *Figure 2*) were performed with 15–30 individuals in a 10 × 10 × 10 cm cage. Groups of mosquitoes were recorded for up to 1 hr and replaced by a new group for a subsequent recording (mosquitoes were not re-used and discarded after experiments). We noticed that activity is typically highest in the first 15–30 min of an experiment, depending on the question being addressed multiple short experiments may therefore yield more data compared to a single long experiment. Individual *Ae. albopictus* females (*Figure 3*) were recorded for 10 min per mosquito and discarded after the experiment. Movement status (*Figure 3A and B*) was classified using the velocity derived from tracking.

### DEET experiments

As DEET dissolves Parafilm and plastics, a glass surface was placed on top of the heated culture flask (no artificial meal was present during DEET experiments). The glass surface was partly coated with 50% N,N-diethyl-meta-toluamide (DEET) using a cotton swab. Groups of 20 *An. coluzzii* females (14 days old) were released into a 10 × 10 × 10 cm cage with the DEET-coated substrate mounted in the floor. Images were acquired at 40 frames per second for 1 hr. Mosquito and body part tracking was performed as described above. The landing rate was calculated by summing the number of trajectories that started on the surface in question (DEET coated versus non-coated) and normalizing this value by the area of the surface. The dwell time was calculated as the average duration of all trajectories on the surface in question. The duration of trajectories moving from the non-coated surface to the DEET-coated surface was split proportionally to the time spend on the respective surface, trajectories moving from the DEET-coated surface to the non-coated surface were not observed indicating that the dwell time on the DEET surface was not limited by the size of the surface.

## Acknowledgements

We thank Emilie Giraud, Gregory Murray, Haripriya Vaidehi Narayanan, Shailabh Kumar, William Gilpin, Hongquan Li, Leslie Vosshall, Veronica Jove, and all members of the Prakash and Lambrechts labs and the Center for Research and Interdisciplinarity for valuable discussions; and Catherine Lallemand, Sylvain Golba, and Patricia Baldacci for help with the rearing of mosquitoes. We thank the reviewers for their constructive comments. *An. stephensi* strain Sda500 and *An. coluzzii* strain N'Gousso were provided by the Centre de Production et Infection d'Anopheles of Institut Pasteur; *Ae. aegypti* strain Liverpool was provided by Leslie Vosshall (Rockefeller University); *Ae. aegypti* strain D2S3 (NR-45838) was provided by Centers for Disease Control and Prevention for distribution by BEI Resources, NIAID, NIH. FJHH was supported by a Rubicon fellowship for the Netherlands Foundation for Scientific Research, a Career Award at the Scientific Interface from the Burroughs Wellcome Fund, and a Marie Curie Fellowship from the European Union. L.L. is supported by the French Agence Nationale de la Recherche (grants ANR-16-CE35-0004-01 and ANR-18-CE35-0003-01), and the French Government's Investissement d'Avenir program Laboratoire d'Excellence Integrative Biology of Emerging Infectious Diseases (grant ANR-10-LABX-62-IBEID). MP was supported by NIH DP2-AI124336 New Innovator Award and USAID Grand Challenges: Zika and Future Threats Award.

## Additional information

### Funding

| Funder | Grant reference number | Author |
| --- | --- | --- |
| Burroughs Wellcome Fund | Career Award at the Scientific Interface | Felix JH Hol |
| H2020 Marie Skłodowska-Curie Actions | PiQMosqBite | Felix JH Hol |
| Nederlandse Organisatie voor Wetenschappelijk Onderzoek | Rubicon | Felix JH Hol |

| Agence Nationale de la Recherche | ANR-16-CE35-0004-01 | Louis Lambrechts |
|---|---|---|
| Agence Nationale de la Recherche | ANR-18-CE35-0003-01 | Louis Lambrechts |
| Agence Nationale de la Recherche | ANR-10-LABX-62-IBEID | Louis Lambrechts |
| National Institutes of Health | DP2-AI124336 | Manu Prakash |
| United States Agency for International Development | Grand Challenges: Zika and Future Threats | Felix JH Hol<br>Manu Prakash |

The funders had no role in study design, data collection and interpretation, or the decision to submit the work for publication.

## Author contributions

Felix JH Hol, Conceptualization, Resources, Data curation, Software, Formal analysis, Funding acquisition, Validation, Investigation, Visualization, Methodology, Writing - original draft, Project administration; Louis Lambrechts, Resources, Project administration, Writing - review and editing; Manu Prakash, Conceptualization, Resources, Funding acquisition, Project administration, Writing - review and editing

## Author ORCIDs

Felix JH Hol (iD) https://orcid.org/0000-0001-8061-0826
Louis Lambrechts (iD) http://orcid.org/0000-0001-5958-2138
Manu Prakash (iD) http://orcid.org/0000-0002-8046-8388

## Decision letter and Author response

Decision letter https://doi.org/10.7554/eLife.56829.sa1
Author response https://doi.org/10.7554/eLife.56829.sa2

# Additional files

## Supplementary files

• Transparent reporting form

## Data availability

Source data files for Figures 2 and 3 are provided as Supplementary Files, code to generate figures is available from Github (under GNU GPLv3 license): https://github.com/felixhol/biteOscope (copy archived at https://github.com/elifesciences-publications/biteOscope).

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

# Appendix 1

**Appendix 1—table 1.** BiteOscope parts list.

The left columns describe the set-up as used for all experiments described in the paper, the right columns describe a low-cost alternative. Vendors are suggestions, costs are in US dollars and approximate.

| Item | Vendor | Part | Cost | Vendor | Part | Cost |
|---|---|---|---|---|---|---|
| | Current | | | Low-cost | | |
| General | | | | | | |
| DC power supply | Instek | GPD-3303D | 400 | home built | | 25 |
| linux computer | intel | NUC7i5BNK | 350 | NVIDIA | Jetson Nano | 100 |
| SSD | samsung | 970 Evo 500 GB | 180 | ScanDisk | | 100 |
| RAM | crucial | 16 GB | 100 | | | |
| Illumination | | | | | | |
| LED array (bright field) | amazon | vidpro LED 312 | 80 | | | |
| LED array (IR) | Taobao | 840/950 nm | 80 | Taobao | 840/950 nm | 80 |
| Imaging | | | | | | |
| camera | Basler | acA2040-90um | 1500 | TIS | DMK 37BUX178 | 350 |
| lens | canon | macro lens EF 100 mm | 600 | HIKVISION | MVL-HF3528M-6MP | 125 |
| lens coupler | fotodiox | pro lens mount EOS - c | 30 | | | |
| Environmental control | | | | | | |
| Temp sensor | digikey | ds18b20 | 19 | | | 19 |
| peltier element | adafruit | 1330 | 12 | adafruit | 1330 | 12 |
| Solenoid valve (gas) | adafruit | 997 | 7 | adafruit | 997 | 7 |
| raspberry pi | amazon | | 35 | | | 35 |
| relay switch | amazon | | 6 | | | 6 |
| jumper wires | any | | 8 | | | 8 |
| Half size bread board | any | | 3 | | | 3 |
| alligator clips | any | | 7 | any | | 7 |
| Bite substrate | | | | | | |
| 70 ml culture flask | Falcon | 353109 | 1 | Falcon | 353109 | 1 |
| PBS | Sigma Aldrich | | 1 | Sigma Aldrich | | 1 |
| ATP | Sigma Aldrich | | 1 | Sigma Aldrich | | 1 |
| parafilm | Sigma Aldrich | | 1 | Sigma Aldrich | | 1 |
| Cage | | | | | | |
| Acrylic sheets | McMaster | Clear Cast | 30 | McMaster | Clear Cast | 30 |
| Total: | | | 3451 | | | 911 |

