## [Decision Letter]

**Acceptance summary:**

This manuscript is of great importance to the field as it represents an important technical advance that will support further studies of mosquito biting and disease-transmitting behavior. By combining image-based tracking, computer vision algorithms, and deep learning, the authors quantify the parameters which are of high relevance to future studies of the neurobiology controlling mosquito blood-feeding and, hence, transmission of human pathogens.

**Decision letter after peer review:**

Thank you for submitting your article "BiteOscope, an open platform to study mosquito blood-feeding behavior" for consideration by *eLife*. Your article has been reviewed by Dominique Soldati-Favre as the Senior Editor, a Reviewing Editor, and three reviewers. The following individuals involved in review of your submission have agreed to reveal their identity: Carlos Ribeiro (Reviewer #2); Philip McCall (Reviewer #3).

The reviewers have discussed the reviews with one another and the Reviewing Editor has drafted this decision to help you prepare a revised submission.

Summary:

The manuscript describes an experimental framework for studying female mosquito foraging and feeding behavior. By combining previously established stimuli promoting female feeding behavior, with image-based tracking, computer vision algorithms, and deep learning, the setup is able to record trajectories from multiple mosquitos within an acrylic box containing an area that contains an artificial meal which the mosquitos can reach by piercing a parafilm membrane. The authors provide data allowing them to quantify different parameters which are of high relevance to future studies of the neurobiology controlling mosquito blood-feeding and hence the potential transfer of pathogens. As proof of principle, they explore the impact of DEET on *An. coluzzii* behavior, which leads them to suggest that chemosensory neurons in the legs but not the proboscis are required for the deterring effect of DEET.

This manuscript is of great importance to the field as it represents an important technical advance that will support further studies of mosquito blood-feeding, a disease-transmitting behavior. The method is super useful, but the technical description and the discussion of the data is a bit superficial. Also, the method is limited to clear liquids, this technical limitation that is difficult to overcome and as such should be acknowledged and discussed in the manuscript. While the different aspects of the method are not novel on their own, the experimental framework is intriguing both in terms of the technical aspects as well as the potential to other researchers in the field. Importantly, the authors make the hardware design as well as the software openly available. The manuscript is well written and concise. However, the reviewers raised some major concerns that will need to be addressed.

The Title:

The mention of blood-feeding in the title is misleading. It should reflect the experimental setup. Therefore, blood-feeding should be replaced by feeding. Temperature is not a good proxy for blood.

Essential revisions:

1) The authors have completely avoided a large literature on the difference between the effects of DEET alone versus DEET and human odor. The authors need to review the literature on this topic more thoroughly and address their interpretation of their data. There is substantial data suggesting that DEET cannot repel mosquitoes in the vapor phase without human odor or other attractive odors. For a review on the topic, the authors should read "The mysterious multi-modal repellency of DEET" (https://doi.org/10.1080/19336934.2015.1079360). It is also untrue that there is evidence presented that Aedes aegypti mosquitoes can sense DEET without human odor in the vapor phase in DeGennaro et al., 2013. The section on DEET needs to be revised to address these issues and those below to fairly describe the authors results in context. DeGennaro et al., 2013 also should be referenced when discussing the separation between contact and olfactory actions of DEET in the mosquito as that was one of the key findings of the publication.

2) It is not clear whether Anopheles mosquitoes are any different that Aedes mosquitoes in regard to the effect of DEET in the vapor phase. There is not enough evidence presented in the paper to come to this conclusion for the reasons listed above.

3) There is some literature that states that ATP is not a phagostimulant in Anopheles species (https://doi.org/10.1111/j.1365-3032.1985.tb00029.x). ATP works well in Aedes species to stimulate blood-feeding behavior. In this manuscript, the authors conclude that ATP has no effect on Anopheline feeding when compared to Aedes aegypti. Key components of the feeding solution are important to induce engorgement, but not the ATP. The authors should provide their arguments about the choice of the feeding solution used in the study place their findings in the context of earlier literature.

4) Another major concern is the lack of description and validation of the behavioral classification methods used in the manuscript. In its current form the authors do not explain how they segment the behavior of the animals into approach/take off, stationary, walking, exploration, engorged etc. The quality of the analysis will largely depend on how well these classifications capture the actual behavior. Likewise, the authors never benchmark their algorithms. It is critical that the authors quantify how often their algorithm misses or wrongly assigns a specific behavior. Given that the quantification of the engorgement volume is a key parameter it would be especially important to focus on that aspect of behavior (e.g. how is, for example, full engorgement defined?). Ideally, the authors would validate the video-based quantification of the ingested volume by measuring the actual ingested volume experimentally. But given the difficulty in performing experiments at the moment, a validation of the video data using manual annotations and acknowledging the limitation in terms of quantifying actual volume should suffice.

5) The authors should also validate and benchmark the performance of the deep learning-based detection of the appendages.

6) The authors mostly analyze movies from experiments with multiple animals. It is widely acknowledged that reliably tracking the identity of multiple animals is challenging. The authors should benchmark their algorithm and provide an error rate for assigning the correct identity to animals. This is key for the correct interpretation of the results.

7) While the use of a membrane to visualize the actual feeding behavior of mosquitoes is a key aspect of the setup, the authors did not fully exploit it. It would be important to go beyond the anecdotal data in the first figure and show analyses of the piercing and stylet behavior highlighting this key aspect of the setup.

8) Some of the statements in the manuscript are rather anecdotal and would be better supported by including their quantification in figures. Furthermore, statistical analysis needs to be described in more details for Figure 3, i.e. include exact p-values in the figure. It also seems that the number of samples (n=9-10) is relatively low for making solid interpretations. Finally, some of the numbers described in the main text do not match the caption label for Figure 2.

9) The quantitative analysis shown in Figure 5 is insufficient, especially because it does not fully support the statements made in the main text. How is the landing rate (and dwell time etc.) calculated? Are these values normalized to the area coated by DEET and inhomogeneities for mosquito landing observed on the arena? Furthermore, the authors should control or at least discuss the possibility that aversiveness is being caused by physical attributes of the coated surface (i.e., slippery surface).

10) The authors' efforts to make the setup openly available including parts descriptions and code repository are highly appreciated. However, reproducibility and openness could be further improved by making the software easier accessible and understandable by structuring the code in the repository and documenting it, because currently, it does not explain which files to use to reproduce the findings. I also could not find the source data of Figure 2 and Figure 3 as described in the data availability statement. Data from all figures should be made available, clearly labeled, code should be provided for reproducing all figures, and well documented for others to use.

11) The Discussion section is rather superficial. A more thorough comparison of how the observed behavior compares to feeding and foraging behavior of other animals, especially insects would be a valuable addition. Also, discussing the limitations of the method would be advisable. The authors should openly recognize and discuss how prudent is an extrapolation of questions around vectorial capacity and host-vector interactions from a minimalist system with synthetic skin, blood, and without human-specific attractants to 'real world'. If the authors believe that it would not be difficult to augment the experimental setup with a human odor (synthetic or real) or any other attractant, then the text should state this clearly.

Revisions expected in follow-up work:

While the current experimental design of the BiteOscope provides advantages to tracking mosquito feeding behavior on humans or animals, a key question which remains unanswered is to which extent the behavior observed on the membrane is comparable to the behavior on a living host. Except for the actual blood feeing behavior, tracking animals foraging on a host should be feasible. It would be an extremely important addition to compare the behavior of mosquitoes in such a naturalistic setting with the behavior on the membrane. Understandably, in the current COVID situation performing experiments is challenging. Therefore, the authors should at least discuss this caveat and consider performing such experiments in follow-up work.

[Editors' note: further revisions were suggested prior to acceptance, as described below.]

Thank you for resubmitting your work entitled "BiteOscope, an open platform to study mosquito blood-feeding behavior" for further consideration by *eLife*. Your revised article has been evaluated by Dominique Soldati-Favre (Senior Editor) and a Reviewing Editor.

The manuscript has been improved but there are some remaining issues that need to be addressed before acceptance, as outlined below:

1) To avoid confusion and false expectations, the title should not include "blood-feeding" but "biting" behavior.

2) The authors should tone down the enthusiasm about the quality of the stylet imaging data in subsection “Automatic characterization of the blood feeding behavior of multiple species” and also mention that using DeepLabcut to track the stylet is not trivial.

3) Please modify the text to clarify the questions of the reviewer 3 regarding responses of the two mosquito species to DEET.

Reviewer #1:

This manuscript presents an exciting new approach to visualizing and characterizing mosquito blood-feeding behavior. This version of the manuscript is substantially revised. It addresses my prior concerns. In particular, I would point to the improved discussion of DEET and how the results presented in this paper fit into our understanding of DEET-mediated repellency. This paper will be of interest to *eLife*'s broad readership and is ready for publication in its current form.

Reviewer #2:

The authors have done a superb job at revising the manuscript and addressing the concerns of the reviewers. Especially given the difficult times.we are all facing. I especially appreciate the thorough validation of the algorithms and the improved description of the methods and the curation of the code on GitHub.

Reviewer #3:

The manuscript is much improved but I'd like some feedback on the DEET story before going any further.

This is a system with many different elements each of which has resolution limits, and the bulk of the reviewers' comments were directed towards getting them recognised and acknowledged. The authors have addressed everything and, in most cases, they seem to have edited and have altered the manuscript sufficiently.

Nonetheless it is ultimately an imaging system and even the best pictures never tell the complete story. For me, a few issues remain.

Blood feeding – given the artificial membrane, the absence of blood/ necessity for clear liquid and presumably subsequent digestion (e.g. peritrophic mem from. Line brane formation?), this is 'biting' behaviour rather than bloodfeeding? This is likely to be relevant to many of the applications listed in the Discussion section.

Similarly, is engorgement an accurate term for what's being measured? Engorgement = fed to repletion, but here that is not always the case and mosquitoes are simply 'fed'.

Also, I wondered whether viewing from directly beneath the ventral abdomen is the most reliable position to measure an abdomen expanding with ingested volume of fluid – i.e. does the abdomen of all individuals expand similarly in every time (e.g. parous vs. nullipars?); what about 3D?

DEET – I found the authors' reply confusing (which read as if Afify and Potter provided more convincing evidence than the authors had.) but the text in the revised manuscript text was much clearer. Nonetheless, I still have reservations: the contact vs. non-contact observations are fine but is this conclusion justified? Can imaging [alone] provide the evidence to solve this question?

1) If the two genera differ in responses to DEET vapour, then in the real world' Anopheles coluzzii would land frequently on DEET-treated skin, whereas Aedes aegypti would rarely/never land. I have no data but having used DEET as a repellent for over 30 years in Africa and elsewhere, I remember Anophelines being repelled completely.

2) In the insecticide world, we use the terms 'contact-irritancy' and 'repellent-induced response', the latter being a change occurring prior to, or without contact. Both are usually bundled together for convenience, often viewed as being a question of exposure dosage from low/vapour to high/contact. I've always had doubts, increasingly so with the recent papers by Ingham et al.

Is it possible that the different responses reported for the 2 genera are the result of different response thresholds, with Aedes being more sensitive at lower levels (vapour) than Anopheles?.… also, have the olfactory neurons in Anopheles coluzzii been explored (which is not mentioned)?

3) Can results from experiments with DEET in the absence of host stimuli be reliable or indicative of anything other than the mosquito can/cannot detect it?

---

## [Author Response]

Summary:The manuscript describes an experimental framework for studying female mosquito foraging and feeding behavior. […] The manuscript is well written and concise. However, the reviewers raised some major concerns that will need to be addressed.

We thank the reviewers and editors for the positive response and are please to read that our work is deemed of great importance to the field. Following the suggestions in the review report we have thoroughly revised our manuscript and now include a more detailed description of our method, and extensive discussion, and we have performed several additional analyses to validate and benchmark our algorithms. We thank the reviewers and editors for the constructive comments.

The Title:The mention of blood-feeding in the title is misleading. It should reflect the experimental setup. Therefore, blood-feeding should be replaced by feeding. Temperature is not a good proxy for blood.

We agree with the reviewers that temperature is not a good proxy for blood. However, we note that full engorgement (the significant swelling of the abdomen when imbibing blood) only occurs when female mosquitoes feed on blood, while on nectar mosquitoes imbibe a much smaller volume (the volume of a typical nectar meal versus a blood meal in Ae. aegypti differs by a factor of 3 to 4 (Jove et al., 2020)). Furthermore, the piercing of a membrane and subsequent insertion of the stylets is a hallmark of blood feeding (Jove et al., 2020). As our method induces these hallmarks of blood feeding (and not nectar feeding) using a liquid that contains the essential components to elicit engorgement (osmotic pressure similar to blood, sodium ions, ATP) and a membrane that the stylets need to pierce to obtain the meal, we feel it is more accurate to refer to it as a method to study blood feeding as opposed to using the more general term ‘feeding’ (which would include e.g. nectar feeding).

Essential revisions:1) The authors have completely avoided a large literature on the difference between the effects of DEET alone versus DEET and human odor. The authors need to review the literature on this topic more thoroughly and address their interpretation of their data. There is substantial data suggesting that DEET cannot repel mosquitoes in the vapor phase without human odor or other attractive odors. For a review on the topic, the authors should read "The mysterious multi-modal repellency of DEET" (https://doi.org/10.1080/19336934.2015.1079360). It is also untrue that there is evidence presented that Aedes aegypti mosquitoes can sense DEET without human odor in the vapor phase in DeGennaro et al., 2013. The section on DEET needs to be revised to address these issues and those below to fairly describe the authors results in context. DeGennaro et al., 2013 also should be referenced when discussing the separation between contact and olfactory actions of DEET in the mosquito as that was one of the key findings of the publication.

We apologize for not discussing the literature on the repellency of DEET clearly. We have revised this part of the Results section thoroughly and have added a section on DEET in the Discussion section to accurately place our observations in the context of the current literature. We now cite DeGennaro et al., (2013) when discussing contact-dependent repulsion of *Ae. aegypti* and have removed the reference to DeGennaro et al., (2013) when discussing the sensing of DEET by *Ae. aegypti* in the absence of human odor, as indeed DeGennaro et al., (2013) do not report on this.

We now furthermore discuss (Discussion section) a recent study comparing olfactory behaviors in response to DEET across several species by Afify and Potter (Afify and Potter, 2020) reporting that *Ae. aegypti* are repelled by DEET in the vapor phase presented in the absence of human odors. In contrast to previous studies that were difficult to interpret because they could not discriminate between olfactory-mediated repulsion and contact-dependent repulsion (e.g. Syed and Leal, (2008)), Afify and Potter, (2020) prevented mosquitoes from coming into contact with DEET to measure a purely olfactory response and observed that volatile DEET (100% concentration, presented alone) repelled *Ae. aegypti* (*An. coluzzii* did not respond to DEET in this assay, while *Cx. Quinquefasciatus* showed stronger repulsion compared to *Ae. aegypti*). We realize that this finding (reported by Afify and Potter, (2020)) contrasts with the statement brought up in the review report: “There is substantial data suggesting that DEET cannot repel mosquitoes in the vapor phase without human odor or other attractive odors”. However, we believe that this discrepancy may reflect the fact that many studies reporting on the effects of DEET do not assay repulsion by volatile DEET per se, yet test the absence of attraction in the presence of volatile DEET when mosquitoes face a choice between attractive odor alone versus attractive odor presented *with* DEET. In addition to Afify and Potter, (2020), we now also cite Stanczyk et al., (2013) which measured attraction of *Ae. aegypti* to a heat source covered with either DEET or solvent impregnated cloth. No human odor was presented during this experiment and the heat source with impregnated cloth was placed outside the mosquito cage (0.5 cm from the mesh) to prevent contact. Stanczyk et al., (2013) observed a strongly reduced attraction of *Ae. aegypti* to the DEET treated heat source, compared to the control heat source treated with solvent (í 30~ attracted to the control, 0% to the DEET treated heat source).

We also note that, in contrast to these two studies reporting repulsion of *Ae. aegypti* by volatile DEET without the presence of attractive odors (Stanczyk et al., 2013; Afify and Potter, 2020), an earlier study (Dogan et al., 1999) reported attraction of *Ae. aegypti* by volatile DEET alone. We now mention this diversity of results in the Discussion section and additionally point to (DeGennaro, 2015) for a review of the multi-modal repellency of DEET.

In summary, to place our observations regarding DEET repellency in context we now discuss:

1) Contact-based repellency by DEET observed in:

- *An. coluzzii* (our results)

- *Ae. aegypti* (DeGennaro et al., 2013; Dennis et al., 2019).

2) Olfactory responses to volatile DEET; we note that not all of these may be exhibited by a mosquito species, and (a subset of) these could potentially act in concert:

- Avoidance of DEET, also in the absence of human odors (the ‘smell and avoid’ hypothesis) e.g. observed in (Stanczyk et al., 2013; Afify and Potter, 2020)

- Inhibition of attraction to human odorants either through modulation of the olfactory system by ‘scrambling of the odor code’ (e.g. the ‘confusant’ hypothesis) and/or ‘smell and avoid’, for example observed in *Ae. aegypti* (DeGennaro et al., 2013)

- ‘Masking’ of odorants by DEET through direct chemical interactions between DEET and odorants, e.g. described in *An. coluzzii* (Afify et al., 2019)

2) It is not clear whether Anopheles mosquitoes are any different that Aedes mosquitoes in regard to the effect of DEET in the vapor phase. There is not enough evidence presented in the paper to come to this conclusion for the reasons listed above.

The DEET experiments described in our manuscript sought to assess if *An. coluzzii* was repelled upon direct contact with DEET (and we conclude this is indeed the case). For effects in the vapor phase we refer to the literature and in response to point #1 have expanded this discussion. We cite a study by Afify and Potter, (2020) that observed a difference between *Ae. aegypti* and *An. Coluzzii* responding to DEET in the vapor phase: *An. coluzzii* was not repelled by 100% DEET whereas *Ae. aegypti* showed moderate repulsion. See Afify and Potter, (2020).

Several studies report the activation of olfactory neurons in the *Ae. aegypti* antennae by volatile DEET presented alone without human body odors (Davis and Rebert, 1972; Boeckh et al., 1996; Stanczyk et al., 2010). Activity-dependent Ca^2+^ imaging in olfactory neurons of *An. coluzzii* on the other hand, suggests that volatile DEET presented alone does not activate olfactory neurons in the *An. coluzzii* antennae (Afify et al., 2019). This indicates that DEET in the vapor phase has a different effect on activation of the olfactory neurons of *Aedes* and *Anopheles* mosquitoes, and the repulsion assay by Afify and Potter, (2020) suggests this may have distinct behavioral effects.

Our experiments focussed on the contact-dependent effects of DEET on *An. coluzzii*, and it is interesting to note that our observations in *An. coluzzii* are very similar to those of DeGennaro et al., (2013) and Dennis et al., (2019) reporting on contact-dependent repulsion in *Ae. aegypti*. In contrast to the similarities in contact-dependent repulsion by DEET, the literature discussed above (and in response to point #1) suggests that the olfactory responses to volatile DEET may to differ to some degree between the two species. To place our results in context, we now discuss these observations extensively in the Results section and the Discussion section.

3) There is some literature that states that ATP is not a phagostimulant in Anopheles species (https://doi.org/10.1111/j.1365-3032.1985.tb00029.x). ATP works well in Aedes species to stimulate blood-feeding behavior. In this manuscript, the authors conclude that ATP has no effect on Anopheline feeding when compared to Aedes aegypti. Key components of the feeding solution are important to induce engorgement, but not the ATP. The authors should provide their arguments about the choice of the feeding solution used in the study place their findings in the context of earlier literature.

We are aware of the study by Galun et al., (1985) demonstrating that ATP is not required to induce engorgement in several *Anopheles* species. However, as we did not perform feeding experiments with *Anopheles* mosquitoes without the presence of ATP, we did not discuss this study in the manuscript. We agree with the reviewers that this could be of interest to the readers and now cite Galun et al., (1985) on line 101 mentioning that ATP may not be required to induce engorgement in *Anopheles*.

All feeding experiments presented in the manuscript were performed using phosphate buffered saline (PBS) with 1 mM ATP as the artificial meal (with the exception of ‘PBS only’ experiments presented in Figure 3). We based our choice for the artificial meal on Galun et al., (1963) and Duvall et al., (2019) which demonstrate that eliciting the feeding response in *Ae. aegypti* requires sodium ions and an osmotic pressure close to blood. Galun et al., (1963) observed the highest percentage of feeding mosquitoes using 10 mM ATP in 150 mM NaCl (the same concentration as typical in PBS) and Duvall et al., (2019) observed robust feeding using 1 mM ATP in 120 mM NaHCO3. We chose 1 mM ATP in PBS as the artificial meal as it is a commonly used buffer that fits the criteria described by Galun et al., (1963) and Duvall et al., (2019). As Galun et al., (1985) report engorgement of *Anopheles* mosquitoes on 150 mM NaCl (neutralized with NaHCO3) that is independent of ATP, we chose for consistency across our experiments and used the same formulation for experiments with *Aedes* and *Anopheles* mosquitoes.

A separate study led by the Vosshall lab in which we collaborated characterizing blood detection in the *Ae. aegypti* stylet (currently under review and available on bioRxiv (Jove et al., 2020)) used the biteOscope and showed that *Ae. aegypti* also engorge on a simplified saline meal consisting of 1 mM ATP in 110 mM NaCl and 20 mM NaHCO3 (compared to PBS: 150 mM NaCl, 1 mM KH2PO4, 3 mM Na2HPO4). We have verified that *Ae. albopictus*, *An. stephensi*, and *An. coluzzii* also feed robustly on this simpli1ed formulation and now mention the possibility to use 1 mM ATP in 110 mM NaCl and 20 mM NaHCO3 in the Materials and methods section.

4) Another major concern is the lack of description and validation of the behavioral classification methods used in the manuscript. In its current form the authors do not explain how they segment the behavior of the animals into approach/take off, stationary, walking, exploration, engorged etc. The quality of the analysis will largely depend on how well these classifications capture the actual behavior. Likewise, the authors never benchmark their algorithms. It is critical that the authors quantify how often their algorithm misses or wrongly assigns a specific behavior. Given that the quantification of the engorgement volume is a key parameter it would be especially important to focus on that aspect of behavior (e.g. how is, for example, full engorgement defined?). Ideally, the authors would validate the video-based quantification of the ingested volume by measuring the actual ingested volume experimentally. But given the difficulty in performing experiments at the moment, a validation of the video data using manual annotations and acknowledging the limitation in terms of quantifying actual volume should suffice.

We agree with the reviewers that adding additional details on the accuracy of the behavioral classification methods is a very useful addition to the manuscript. We performed a thorough benchmarking of our locomotion behavior classification and engorgement detection algorithms and have now included the results of this analysis in the text as indicated below and as Figure 3—figure supplement 1.

Locomotion behavior classification

The classification of locomotion behaviors is based on the velocity derived from centroid tracking. In the original manuscript we mentioned velocity based locomotion classification very briefly in the Discussion section. Following the suggestions of the reviewers, we now include information regarding the performance of our classifier of locomotion behaviors as presented in Figure 3A and B of the manuscript. To quantify performance, we manually labeled the behavior of *Ae. Albopictus* mosquitoes exhibited in 1124 frames of the dataset presented in Figure 3 of the manuscript, and compared the labeled behaviors to the computationally detected behaviors. The overall accuracy of behavioral classification was 89%, with a per class accuracy of 90% (stationary), 89% (walking), and 97% (flight), with accuracy defined as: TP+TNO, with TP denoting true positives, TN true negatives, and O the number of observations. The classification of locomotion behaviors depends on the velocity thresholds set to distinguish flight, walking, and stationary behaviors, we therefore performed a sensitivity analysis to estimate the dependence of the classification accuracy on the thresholds used. Figure 2 shows the results of this analysis, demonstrating that classification accuracy peaks at 89% accurate classifications using a stationary – walking threshold of 2 mm/s and a walking – flight threshold of 12 mm/s, and is superior to 80% accurate for a range of parameters. We now include this information in the Results section and the Materials and methods section, and have included the figure detailing the threshold-dependence of classification accuracy as Figure 3—figure supplement 1.

Engorgement detection

We now included a thorough characterization of the performance of the engorgement detection algorithm, and have included additional details in the Materials and methods section regarding the computational detection of full engorgement. To determine engorgement, we fit an active contour model to the mosquito’s body (with appendages computationally removed) and we use the area of the fitted shape as a proxy for engorgement. As can be seen in Video 7 and panels 1 and 2 of Figure 1H of the manuscript, the abdominal width increases during imbibing and saturates when a mosquito is fully engorged. After full engorgement, mosquitoes retract their mouthparts from the artificial blood meal and often remain stationary for a period to excrete excess liquid. In anophelines this is clearly visible as a growing droplet at the abdominal tip, whereas in *Aedes* this is less pronounced yet occasionally droplet excretion can be seen. For a mosquito to be computationally defined as engorged, two conditions need to be met:

1) The abdominal area needs to expand by 1.3 fold, where fold expansion is calculated as the ratio of the 90th percentile of abdominal area measurements and the 10th percentile of abdominal area measurements in the first 10 seconds of the trajectory.

2) The ninetieth percentile of abdominal area measurements needs to exceed 2.4 mm2 for *An. stephensi* and *An. coluzzii*, or 3.0 mm2 for *Ae. aegypti* and *Ae. albopictus*.

Following the reviewers’ suggestion (as indeed our ability to do experiments is severely limited during the Covid-19 crisis), we estimated the performance of the engorgement detection algorithm by validating all data presented in Figure 2 and Figure 2—figure supplement 1. Visual inspection of all video data indicated that the overall sensitivity of engorgement detection TPp=0.81(n=130), with a sensitivity of 0.97 (*n* = 29) and 0.76 (*n* = 101) for *Aedes* and *Anopheles* mosquitoes, respectively. The overall specificity was TNN=1.0(n=101). We now include these numbers in the manuscript on lines 136-137 and details regarding the benchmarking in subsection “Detecting engorgement”.

We note that the sensitivity for detecting engorgement is lower for *Anopheles* compared to *Aedes* mosquitoes (0.76 versus 0.97). Two possible reasons may be the source of this discrepancy: (1) *Anopheles* excrete excess liquid during feeding to a much larger extent than *Aedes* mosquitoes, resulting in a less pronounced dilation of the abdomen making it harder to detect the dilation, (2) we noticed a higher density of mosquitoes on the bite substrate in experiments conducted with *Anopheles* compared to *Aedes* mosquitoes, the higher density more often leads to mosquitoes that physically touch resulting in challenging situations for the detection algorithm as the active contour model may fit the body shape less accurate when mosquitoes touch. Figure 3 shows the relation between the average number of mosquitoes present on the bite substrate versus the accuracy of engorgement detection. A lower number of mosquitoes on the substrate results in higher accuracy, yet at similar mosquito densities the algorithm performs better on *Aedes* than on *Anopheles*, likely due to a larger increase in abdominal area during feeding in *Aedes*.

5) The authors should also validate and benchmark the performance of the deep learning-based detection of the appendages.

In the original manuscript we described the accuracy of the deep learning-based tracking of appendages in subsection “Pose estimation, behavioral classification, and contact-dependent sensing”, and the Discussion section stating an average accuracy of the detection of body parts of 10 pixels (250 *µ*m) and 8 pixels (200 *µ*m) for *Aedes* and *Anopheles*, respectively (average distance between manually labeled and computationally predicted body part location). We now include additional benchmarking and details regarding the performance of deep learning-based body part detection in subsection “Body part tracking”.

In order to assess the robustness of the trained model, we created multiple shuffles of the training data set. ‘Shuffle’ here refers to the generation of a training set from a pool of manually labeled images: labeled images are randomly split into a ‘train’ and a ‘test’ set, the random split differs per shuffle allowing one to assess the ‘robustness’ of training.

Below we report the accuracy of body part detection defined as the mean average Euclidean error between the manual labels and the ones predicted by the algorithm. Averaged over 4 shuffles of the training set *Aedes* body parts were detected with a mean accuracy of 11 pixels, (275 *µ*m). The mean accuracy of predicting *Anopheles* body parts averaged over 2 shuffles of the training set was 6 pixels (150 *µ*m). In addition to the mean performance across all body parts, we now also include accuracies per groups of body parts for the best performing model (core: head, proboscis, abdomen, abdominal tip, (and palps for *Anopheles*); and legs: tips of all 6 legs). In the best performing model for detecting *Aedes* body parts the mean distance between manual labels and predicted positions was 1.7 pixels (43 *µ*m) for core body parts, and 1.6 (40 *µ*m) for the tips of legs. The best performing model for *Anopheles* body parts had an accuracy of 5.2 pixels (130 *µ*m) and 3.7 pixels (93 *µ*m) for core and legs, respectively. We now include these metrics in the manuscript in subsection “Body part tracking” (and briefly in the Results section). We furthermore provide the final trained models on Github https://github.com/felixhol/biteOscope (mentioned subsection “Body part tracking”). Depending on the exact imaging conditions of users, these trained models may work without modification on newly acquired data, in other cases the provided models can be used as a convenient starting point for training on new datasets. In the latter case, starting from the pre-trained models will reduce the required training time substantially making training on a standard computer without GPU feasible.

6) The authors mostly analyze movies from experiments with multiple animals. It is widely acknowledged that reliably tracking the identity of multiple animals is challenging. The authors should benchmark their algorithm and provide an error rate for assigning the correct identity to animals. This is key for the correct interpretation of the results.

We agree with the reviewers that correctly tracking the identity of multiple mosquitoes that are simultaneously present on the bite substrate is an important aspect of our image analysis pipeline. It indeed is a great suggestion to present the validation in the manuscript.

In order to assess the performance of our mosquito tracking algorithm, we created videos in which the results of the tracking algorithm are overlaid on the raw imaging data. The validation videos indicate the location where the centroid of a mosquito is detected and plot the numeric ID assigned to a mosquito. We verified the tracking results of 111 individual trajectories across 12293 images resulting in an error rate of 0.045 (5/111). The validation dataset includes data from both *Aedes* and *Anopheles* experiments and consists of images having a variety of densities ranging from 0.05 – 0.4 mosquitoes per cm2. The most common error (4/5) is caused by erroneously assigning identities when mosquitoes cross (e.g. an individual moving over another one and thus overlapping in the image). Interestingly, the validation videos make it straightforward to correct ID swap errors by manually re-assigning the correct identity to the track. A rather minor amount of manual interventions therefore results in nearly perfect tracking. We now include information regarding the performance of the tracking algorithm in subsection “Automatic characterization of the blood feeding behavior of multiple species” and Subsection “Computaional tools”. We include an example validation video, Video 1—figure 1, and provide code to create the validation videos in the Github repository.

7) While the use of a membrane to visualize the actual feeding behavior of mosquitoes is a key aspect of the setup, the authors did not fully exploit it. It would be important to go beyond the anecdotal data in the first figure and show analyses of the piercing and stylet behavior highlighting this key aspect of the setup.

We agree with the reviewers that visualizing the stylet of a probing/feeding mosquito presents very interesting opportunities. This imaging capability for instance was of value to one of the ‘early adopters’ of the biteOscope (the lab of Leslie Vosshall) in a study that characterizes the sensory perception of blood in *Ae. aegypti* mouthparts. In this study mosquitoes were presented membrane feeders containing a variety of artificial meals and the question was raised whether the stylet came into contact with all meals presented, or only with those that elicited engorgement (i.e. are non-appetitive meals rejected based on evaluation by the stylet, or by other means). We used the biteOscope to answer this question and demonstrated that the stylet indeed came in contact with all meals and evaluates the meal before engorging. This study in which we collaborated is currently under review and available on bioRxiv (Jove et al., 2020). We now discuss this use-case of stylet imaging in the Discussion section.

We believe that there are many more questions that can be addressed by directly visualizing the stylet of a feeding mosquito and anticipate that computationally detecting stylet insertion would be a very valuable tool. As the reviewers note’ it would be convenient to use DeepLabCut to detect the stylet. We have indeed tried this however, our efforts have not produced an algorithm that detects the stylet with high enough accuracy to be useful. The reason for this is (at least) two fold: (1) the stylet appears as a rather subtle feature in images, having low contrast (compared to other body parts) and adopting a variety of conformations complicating reliable detection; and (2) the stylet is only visible during probing and feeding and therefore only appears in a minor subset of images resulting in a low number of training images (and a high rate of false positive detections at the tip of the proboscis). We are currently exploring other deep learning methods (not based on DeepLabCut) for stylet detection and hope that this will result in a high-confidence stylet detection algorithm. Given the challenging nature of this problem, and the range of (computational) capabilities we currently present, we feel that high-accuracy computational detection of the stylet is beyond the scope of the current work.

In addition to a number of behavioral statistics, our current computational pipeline saves cropped videos of all individually tracked mosquitoes and we mention in subsection “Computational tools” that these videos may serve as a convenient starting point for manual annotation of stylet piercing as has been done in Jove et al., (2020).

8) Some of the statements in the manuscript are rather anecdotal and would be better supported by including their quantification in figures. Furthermore, statistical analysis needs to be described in more details for Figure 3, i.e. include exact p-values in the figure. It also seems that the number of samples (n=9-10) is relatively low for making solid interpretations. Finally, some of the numbers described in the main text do not match the caption label for Figure 2.

In response to the points raised above, we have performed several analyses (see points 4–6 above) and now include additional quantification in the manuscript. Following the reviewers’ suggestion, we now include the exact *p*-values in the caption of Figure 3. We furthermore noticed that some data points were accidentally excluded from panel 3 of Figure 3C (bout length), we have updated this graph to include all data points. The conclusions regarding this Figure are not affected by this. We agree that using a larger number of individual mosquitoes (9–10 for the data presented in Figure 3) would give our statistical analysis more power. However, the differences between the two experimental conditions are large enough to observe statistically significant differences between the treatments with the current number of samples. Furthermore, we discuss the data from Figure 3 (single mosquito experiments) in relation to the results from Figure 2E which have a much larger *n* (population experiments, *n* = 111 trajectories for *Ae. albopictus*, *n* = 1184 trajectories for all species together) and note that these different data sets are in agreement. It will be interesting to follow up on these findings at a larger scale once we can resume experiments post Covid-19. We thank the reviewers for pointing out a typo in the *n* quoted in the text discussing Figure 2. The caption of Figure 2 indeed specified *n* = 349, whereas the main text read *n* = 350 trajectories for the *An. coluzzi* experiment. The correct number is *n* = 349 and we have corrected this.

9) The quantitative analysis shown in Figure 5 is insufficient, especially because it does not fully support the statements made in the main text. How is the landing rate (and dwell time etc.) calculated? Are these values normalized to the area coated by DEET and inhomogeneities for mosquito landing observed on the arena? Furthermore, the authors should control or at least discuss the possibility that aversiveness is being caused by physical attributes of the coated surface (i.e., slippery surface).

The landing rate was calculated by summing the number of trajectories that started on the surface in question (DEET coated versus non-coated) and normalizing this value by the area of the surface (we now explicitly mention normalization by the surface area in subsection “DEET repels *An. coluzzii* upon contact with legs”). The dwell time was calculated as the average duration of all trajectories on the surface in question. The duration of trajectories moving from the non-coated surface to the DEET coated surface was split proportionally to the time spent on the respective surface. Trajectories moving from the DEET coated surface to the non-coated surface were not observed indicating that the dwell time on the DEET surface was not limited by the size of the surface. We now include this information in subsection “DEET experiments”. Following the reviewers’ suggestion we now also discuss the possibility that physical attributes of the DEET coating may influence the mosquitoes’ interaction with the surface in the Discussion section.

10) The authors' efforts to make the setup openly available including parts descriptions and code repository are highly appreciated. However, reproducibility and openness could be further improved by making the software easier accessible and understandable by structuring the code in the repository and documenting it, because currently, it does not explain which files to use to reproduce the findings. I also could not find the source data of Figure 2 and Figure 3 as described in the data availability statement. Data from all figures should be made available, clearly labeled, code should be provided for reproducing all figures, and well documented for others to use.

We agree with the reviewers that making our setup and code openly available is important, and the reviewers are right in pointing out that the ‘user friendliness’ of the Github repository at the time of initial submission could be improved. We have now restructured the repository to make it easier to navigate. In the repository, we provide a README file that specifies the function of each algorithm and we have commented the code extensively indicating the function of code blocks and variables that should be modified to provide experiment specific details (e.g. directories where data is stored). We furthermore provide test data in the repository to enable testing of the code by new users. In addition to these improvements with respect to accessibility of the code, we now also provide design files for laser cutting the mosquito cages. Data will be available on Github upon acceptance of the manuscript.

11) The Discussion section is rather superficial. A more thorough comparison of how the observed behavior compares to feeding and foraging behavior of other animals, especially insects would be a valuable addition. Also, discussing the limitations of the method would be advisable. The authors should openly recognize and discuss how prudent is an extrapolation of questions around vectorial capacity and host-vector interactions from a minimalist system with synthetic skin, blood, and without human-specific attractants to 'real world'. If the authors believe that it would not be difficult to augment the experimental setup with a human odor (synthetic or real) or any other attractant, then the text should state this clearly.

Following these suggestions, we have thoroughly revised the Discussion section. We now include a more extensive discussion of feeding and foraging behavior in other animals and methods to study those behaviors in the Discussion section. In addition, we point out several interesting opportunities for future research where the biteOscope could be a useful tool (Discussion section). We furthermore discuss several limitations of our setup and the interesting possibility to add odorants to the setup to more closely mimic a real host (Discussion section). We note that a synthetic bite substrate will likely never by exactly as attractive as a real host, yet we also that many factors that may change behavior (e.g. infections/nutritional status or components of the microbiome) are best assessed in a relative manner, e.g. comparing non-infected to infected individuals. We indicate that comparing cohorts of mosquitoes undergoing different experimental treatments puts less emphasis on the absolute attractiveness of the bite substrate and thus mitigates potential issues related to the fact that the a synthetic bite substrate is likely less attractive than a real live host. We discuss the potential use of non-clear liquids in the Discussion section.

Revisions expected in follow-up work:While the current experimental design of the BiteOscope provides advantages to tracking mosquito feeding behavior on humans or animals, a key question which remains unanswered is to which extent the behavior observed on the membrane is comparable to the behavior on a living host. Except for the actual blood feeing behavior, tracking animals foraging on a host should be feasible. It would be an extremely important addition to compare the behavior of mosquitoes in such a naturalistic setting with the behavior on the membrane. Understandably, in the current COVID situation performing experiments is challenging. Therefore, the authors should at least discuss this caveat and consider performing such experiments in follow-up work.

As mentioned above, we now briefly discuss the comparison of behavior in our synthetic system to behavior on a real living host in the Discussion section. We furthermore agree with the reviewers that this would be an interesting follow up and will explore this in future work (when Covid-19 related limitations are lifted).

[Editors' note: further revisions were suggested prior to acceptance, as described below.]

The manuscript has been improved but there are some remaining issues that need to be addressed before acceptance, as outlined below:1) To avoid confusion and false expectations, the title should not include "blood-feeding" but "biting" behavior.

We have changed the title accordingly, it now reads “BiteOscope, an open platform to study mosquito biting behavior”.

2) The authors should tone down the enthusiasm about the quality of the stylet imaging data in subsection “Automatic characterization of the blood feeding behavior of multiple species” and also mention that using DeepLabcut to track the stylet is not trivial.

We have toned down subsection “Automatic characterization of the blood feeding behavior of multiple species” and now mention that tracking the stylet using DeepLabCut is not trivial in subsection “Pose estimation, behavioral classification, and contact-dependent sensing”.

3) Please modify the text to clarify the questions of the reviewer 3 regarding responses of the two mosquito species to DEET.

We respond to the questions of reviewer #3 below and have revised the text accordingly.

Reviewer #1:This manuscript presents an exciting new approach to visualizing and characterizing mosquito blood-feeding behavior. This version of the manuscript is substantially revised. It addresses my prior concerns. In particular, I would point to the improved discussion of DEET and how the results presented in this paper fit into our understanding of DEET-mediated repellency. This paper will be of interest to eLife's broad readership and is ready for publication in its current form.

We thank the reviewer for their support and we are glad that the reviewer appreciates our improved discussion on DEET-mediated repellency.

Reviewer #2:The authors have done a superb job at revising the manuscript and addressing the concerns of the reviewers. Especially given the difficult times we are all facing. I especially appreciate the thorough validation of the algorithms and the improved description of the methods and the curation of the code on GitHub.

We thank the reviewer for their support and we are glad that the reviewer appreciates the thorough validation of our algorithms and our efforts at making our code (available at GitHub) more user friendly.

Reviewer #3:The manuscript is much improved but I'd like some feedback on the DEET story before going any further.

We are pleased to read that the reviewer evaluates our manuscript as much improved and we provide feedback regarding the DEET story below.

This is a system with many different elements each of which has resolution limits, and the bulk of the reviewers' comments were directed towards getting them recognised and acknowledged. The authors have addressed everything and, in most cases,, they seem to have edited and have altered the manuscript sufficiently.Nonetheless it is ultimately an imaging system and even the best pictures never tell the complete story. For me, a few issues remain.Blood feeding – given the artificial membrane, the absence of blood/ necessity for clear liquid and presumably subsequent digestion (e.g. peritrophic mem from. Line brane formation?), this is 'biting' behaviour rather than bloodfeeding? This is likely to be relevant to many of the applications listed in the Discussion section.

Following this comment (and the Editor’s suggestion) we have changed the title of the manuscript which now states ‘biting behavior’ instead of ‘blood feeding behavior’.

Similarly, is engorgement an accurate term for what's being measured? Engorgement = fed to repletion, but here that is not always the case and mosquitoes are simply 'fed'.Also, I wondered whether viewing from directly beneath the ventral abdomen is the most reliable position to measure an abdomen expanding with ingested volume of fluid – i.e. does the abdomen of all individuals expand similarly in every time (e.g. parous vs. nullipars?); what about 3D?

We presented a validation of our engorgement detection algorithm in the first revision of our manuscript, and as suggested by the reviewers, we used manual annotations to validate all video data and calculate the sensitivity (0.97 for *Aedes* and 0.76 for *Anopheles*) and specificity (1.0 for both) of engorgement detection. This validation was mentioned in subsection “Automatic characterization of the blood feeding behavior of multiple species”, and subsection “Detecting engorgement”. While 3D imaging and/or adding another camera (e.g. a side view) could add another way of assessing engorgement status, we feel that this would significantly complicate the optical set up while the sensitivity and specificity of our detection algorithm are already quite high. We furthermore note that the artificial meal we use (phosphate buffered saline + 1 mM ATP) has been used to study blood feeding in several other studies. Notably, Duvall et al., (2019) and Jove et al., (2020) compare the weight of *Ae. aegypti* mosquitoes fed on sheep blood to the weight of mosquitoes fed on saline + ATP and both studies report no significant difference in the post-feeding weight of mosquitoes feeding on either meal (please see Figure 1C of (Duvall et al., 2019) and Supplemental Figure S1F and H of (Jove et al., 2020)). As weight is a good proxy for ingested volume, these observations indicate that there is no significant difference between the ingested volume when using sheep blood versus our artificial meal.

DEET – I found the authors' reply confusing (which read as if Afify and Potter provided more convincing evidence than the authors had.) but the text in the revised manuscript text was much clearer. Nonetheless, I still have reservations: the contact vs. non-contact observations are fine but is this conclusion justified? Can imaging [alone] provide the evidence to solve this question?

We apologize if our response was confusing, yet are glad to read that the manuscript text is clear. We would like to stress that we report on a different aspect of DEET repellency than Afify and Potter did: We tested if *An. coluzzii* is repelled upon *contact* with DEET (a hypothesis not tested before), whereas Afify and Potter studied olfactory responses to *volatile* DEET (i.e. responses in the *absence* of contact). As we study a different aspect of DEET-mediated repellency, our results are complimentary to, instead of more/less convincing than, those of Afify and Potter. In the Discussion section we mention studies by Afify and Potter, and others to place our observations regarding contact-dependent repellency in the broader context of the several modes in which DEET may repel mosquitoes. We extensively discuss our results concerning DEET in relation to the prior literature as this was an explicit request in the initial review report.

1) If the two genera differ in responses to DEET vapour, then in the real world' Anopheles coluzzii would land frequently on DEET-treated skin, whereas Aedes aegypti would rarely/never land. I have no data but having used DEET as a repellent for over 30 years in Africa and elsewhere, I remember Anophelines being repelled completely.

This is an interesting observation which may be explained by the various possible modes of action of DEET (these different modes of action are discussed in the Discussion section). Our results show that *An. coluzzii* is repelled by DEET upon contact, and DeGennaro et al., (2013) and Dennis et al., (2019) report that this is also the case in Ae. aegypti. Regarding olfactory responses to volatile DEET in the absence of contact, several studies suggest that the olfactory response to volatile DEET of these two species differs (Davis and Rebert, 1972; Boeckh et al., 1996; Stanczyk et al., 2010, 2013; Afify et al., 2019) which may result in differences in olfactory behavioral responses to volatile DEET: *Ae. aegypti* has been observed to avoid volatile DEET in the absence of attractive cues, whereas *An. coluzzii* does not seem to avoid volatile DEET in the absence of attractive cues (Afify and Potter, 2020). A third mode of action of DEET is “masking” in which DEET acts directly on the odorants emanating from a host: through chemical interactions DEET decreases the odorants volatility and thereby reduces the amount of attractive odorants capable of activating mosquito olfactory receptors and thus inhibiting behavioral responses (Syed and Leal, 2008). A recent study (Afify et al., 2019) observed that the olfactory neurons of *An. coluzzii* are not activated by DEET, and also showed that the neuronal response to otherwise attractive compounds (e.g. 1-octen-3-ol) was strongly inhibited when the volatile attractive compound was co-presented with volatile DEET (i.e. compound that normally elicit a strong response in olfactory neurons did not elicit a strong neuronal response when presented together with volatile DEET). This observation suggests that DEET interacts directly with odorants and through masking indeed may inhibit behavioral responses (such as flying towards a host). Together, our results and prior literature thus suggest that DEET has (at least) two effects on *An. coluzzii*: contact-based repellency, and the masking of odorants. It is thus likely that DEET has kept reviewer #3 safe from anopheline bites through the combined effect of odorant masking and contact-dependent repulsion.

2) In the insecticide world, we use the terms 'contact-irritancy' and 'repellent-induced response', the latter being a change occurring prior to, or without contact. Both are usually bundled together for convenience, often viewed as being a question of exposure dosage from low/vapour to high/contact. I've always had doubts, increasingly so with the recent papers by Ingham et al.Is it possible that the different responses reported for the 2 genera are the result of different response thresholds, with Aedes being more sensitive at lower levels (vapour) than Anopheles?.… also, have the olfactory neurons in Anopheles coluzzii been explored (which is not mentioned)?

This is an interesting suggestion, yet as our DEET assay was designed to primarily test for contact dependent behaviors, our data do not allow us to draw conclusions regarding the concentration dependency of olfactory (and other non-contact) behaviors. As mentioned above, prior literature suggests that at the same concentration volatile DEET has distinct effects on the olfactory neurons of *Ae. aegypti* and *An. coluzzii*.

Following the reviewers suggestion we now mention that these effects may be concentration dependent in the Discussion section.

The olfactory neurons of *An. coluzzii* have indeed been studied. It may be a misunderstanding that this was not mentioned as we discussed studies reporting on the response of the olfactory neurons of *An. coluzzii* to DEET in subsection “DEET repels *An. coluzzii* upon contact with legs” and Discussion section. In summary, prior literature shows that the olfactory neurons of *Ae. aegypti* are activated by DEET (Davis and Rebert, 1972; Boeckh et al., 1996; Stanczyk et al., 2010) whereas the olfactory neurons of *An. coluzzii* are not activated by DEET (Afify et al., 2019; Afify and Potter, 2020).

3) Can results from experiments with DEET in the absence of host stimuli be reliable or indicative of anything other than the mosquito can/cannot detect it?

This is an interesting question. However, we note that in our assay we do provide heat which is an important factor in short-range host attraction. Our results are therefore best interpreted in the presence of a host stimulus (heat).